# The RNF214-TEAD-YAP signaling axis promotes hepatocellular carcinoma progression via TEAD ubiquitylation

Mengjia Lin [1,2], Xiaoyun Zheng [2,3], Jianing Yan [4], Fei Huang [2], Yilin Chen [2,3], Ran Ding [2,3], Jinkai Wan [5,6], Lei Zhang [5], Chenliang Wang [2], Jinchang Pan [2], Xiaolei Cao [2,3], Kaiyi Fu [2], Yan Lou [7], Xin-Hua Feng [2,3,8], Junfang Ji [2,3,8], Bin Zhao [2,3,8], Fei Lan [5,6], Li Shen [2,9], Xianglei He [10], Yunqing Qiu [1,7] ✉ & Jianping Jin [2,3,7,8] ✉

RNF214 is an understudied ubiquitin ligase with little knowledge of its biological functions or protein substrates. Here we show that the TEAD transcription factors in the Hippo pathway are substrates of RNF214. RNF214 induces non-proteolytic ubiquitylation at a conserved lysine residue of TEADs, enhances interactions between TEADs and YAP, and promotes transactivation of the downstream genes of the Hippo signaling. Moreover, YAP and TAZ could bind polyubiquitin chains, implying the underlying mechanisms by which RNF214 regulates the Hippo pathway. Furthermore, RNF214 is overexpressed in hepatocellular carcinoma (HCC) and inversely correlates with differentiation status and patient survival. Consistently, RNF214 promotes tumor cell proliferation, migration, and invasion, and HCC tumorigenesis in mice. Collectively, our data reveal RNF214 as a critical component in the Hippo pathway by forming a signaling axis of RNF214-TEAD-YAP and suggest that RNF214 is an oncogene of HCC and could be a potential drug target of HCC therapy.

Ubiquitin is a small signaling protein that can be conjugated to its protein substrates. This process, so called ubiquitylation or ubiquitination, is one of the major protein posttranslational modifications in eukaryotes. The ubiquitylation reaction is sequentially catalyzed by a ubiquitin activating enzyme (E1), a ubiquitin conjugating enzyme (E2) and a ubiquitin ligase (E3)[1]. Thus far, at least 43948 ubiquitylation sites from over 14692 ubiquitylated proteins have been detected experimentally in humans[2], implying that ubiquitylation is related to complex biological processes.

[1]State Key Laboratory for Diagnosis and Treatment of Infectious Disease, and National Clinical Research Center for Infectious Diseases, The First Affiliated Hospital, College of Medicine, Zhejiang University, Hangzhou 310000 Zhejiang, China. [2]Life Sciences Institute, Zhejiang University, Hangzhou 310058 Zhejiang, China. [3]Cancer Center, Zhejiang University, Hangzhou 310058 Zhejiang, China. [4]Department of General Surgery, Sir Run Run Shaw Hospital, Zhejiang University School of Medicine, Hangzhou 310016 Zhejiang, China. [5]International Co-laboratory of Medical Epigenetics and Metabolism of Ministry of Science and Technology, and Shanghai Key Laboratory of Medical Epigenetics, Institutes of Biomedical Sciences, Fudan University, Shanghai 200032, China. [6]Key Laboratory of Carcinogenesis and Cancer Invasion of Ministry of Education, and Liver Cancer Institute, Zhongshan Hospital, Fudan University, Shanghai 200032, China. [7]Zhejiang Provincial Key Laboratory for Drug Clinical Research and Evaluation, Department of Clinical Pharmacy, The First Affiliated Hospital, College of Medicine, Zhejiang University, Hangzhou 310000 Zhejiang, China. [8]Center for Life Sciences, Shaoxing Institute, Zhejiang University, Shaoxing 321000, China. [9]Department of Orthopedics Surgery, School of Medicine, The Second Affiliated Hospital, Zhejiang University, Hangzhou 310009 Zhejiang, China. [10]Department of Pathology, Zhejiang Provincial People's Hospital, Hangzhou 3100014 Zhejiang, China. ✉e-mail: qiuyq@zju.edu.cn; jianping_jin@zju.edu.cn

The ubiquitin machinery can conjugate not only a single ubiquitin, but also polyubiquitin chains on its target proteins. Polyubiquitin chains are synthesized by forming an isopeptide bond between the C-terminal glycine residue in the donor ubiquitin (Gly76) and the ε-amino group of a lysine residue or the amino group of the N-terminal methionine residue in the acceptor ubiquitin[1]. Ubiquitin contains seven lysine residues, therefore, at least eight kinds of polyubiquitin chains could be synthesized by the ubiquitylation machinery. Moreover, mixed and branched polyubiquitin chains have been reported as well[1]. The specific linkages of polyubiquitin chains function as "ubiquitin codes" to determine diverse outcomes of ubiquitylated substrates[3]. Certain polyubiquitin chains, such as lysine-11 (K11), lysine-48 (K48) and branched ones often drive protein degradation via the 26S proteasome[1]. Indeed, the ubiquitin-proteasome system is the major cellular machinery selectively degrading short-lived or unwanted proteins in eukaryotic cells. Howbeit, proteolysis is not the only fate of ubiquitylated proteins. The polyubiquitin chains conjugated via the lysine-63 (K63) or N-terminal methionine (M1) residue of ubiquitin are usually not signals for protein turnover[1]. Instead, these non-proteolytic polyubiquitin chains often cause localization changes or functional alterations of protein substrates. Because of the complexity of ubiquitin codes, protein ubiquitylation regulates virtually every aspect of cellular activities and human health. Dysregulation of ubiquitylation is often linked to many human diseases, including cancer, autoimmune, neurodegenerative and viral diseases[4–6].

Ubiquitylation is a very specific process and the substrate specificity is mainly determined by ubiquitin ligases. Human genome encodes over 600 ubiquitin ligases which contains either a RING finger or a HECT domain[7]. They could ubiquitylate many human proteins, including components of various signaling pathways[2,6,8].

The Hippo pathway was first discovered in *Drosophila melanogaster* via genetic screens to look for genetic mutations leading to overgrowth phenotypes[9,10]. Nowadays, it is clear that the key components of the Hippo pathway are highly conserved from *Drosophila* to human, including MST1/2 and LATS1/2, two pairs of upstream kinases; YAP and TAZ, two downstream effectors and transcription coactivators; and the TEAD family of transcription factors[9,11]. These core players and additional factors coordinate with each other to regulate the transcription of Hippo target genes which controls organ size, cell proliferation, survival, and pathophysiological events[12–17]. The Hippo pathway is also mediated by several types of protein modifications, including acetylation, methylation, phosphorylation, and ubiquitylation[18–25]. Protein ubiquitylation has been shown to control the Hippo pathway by regulating protein stability or changing localization of several core proteins, such as YAP/TAZ, LATS1/2, MOB1, and MST1/2[26–31], but no evidence has been found that ubiquitylation can regulate transcription activities of the Hippo-related transcription factors, and no biological significance of ubiquitylation has been characterized for TEAD proteins, although a few ubiquitylation sites were identified in TEAD1, TEAD2, and TEAD4[32,33].

Here we report that RNF214, a RING finger-containing ubiquitin ligase whose biological functions were poorly characterized, ubiquitylates the TEAD transcription factors at their C-terminal YAP binding domains (YBD) without affecting their protein stability or localization. Instead, RNF214 enhances the interactions between TEADs and YAP/TAZ via the recognition of polyubiquitin chains by YAP/TAZ, therefore increasing the expression levels of Hippo target genes mediated by YAP and TEADs. Moreover, we find that RNF214 is overexpressed in hepatocellular carcinoma (HCC) and promotes HCC tumorigenesis via the Hippo pathway as an oncogene. Our work uncovers a critical mechanism regulating the downstream transcription network of the Hippo pathway by formation of a unique RNF214-TEAD-YAP signaling axis.

## Results

### RNF214 interacts with the TEAD transcription factors

RNF214 is a ubiquitin ligase of the RING finger family and an understudied protein whose biological roles were less characterized. RNF214 is localized in both the cytoplasm and the nucleus (Supplementary Fig. 1a). To figure out the functions of RNF214, we created *Rnf214* knockout mice and generated *Rnf214* knockout (*Rnf214⁻/⁻*) mouse embryonic fibroblast cells (MEFs). CCK8 assays showcased that both the *Rnf214⁻/⁻* and the *Rnf214⁺/⁻* MEFs proliferated significantly slower than the *Rnf214⁺/⁺* MEFs, whereas the *Rnf214⁻/⁻* MEFs grew the slowest (Fig. 1a). These results implied the function of RNF214 is probably associated with cell proliferative processes.

To determine biological functions of RNF214, it is critical to identify its interacting proteins and ubiquitylation substrates at first. Considering that ubiquitin ligases usually stay with their protein substrates transiently, we established an APEX proximity labeling strategy coupled with mass spectrometry[34,35] to identify interacting proteins of RNF214. In this approach, we first fused an engineered ascorbate peroxidase (APEX2) to either N-terminus or C-terminus of RNF214, expressed these two fusion proteins in HLF, an HCC cell line, near the endogenous level (Supplementary Fig. 1b, c), and generated short-lived radicals around the APEX2-RNF214 fusion proteins to label biotin on nearby interactive proteins by adding hydrogen peroxide ($H_2O_2$) and biotin-phenol (also called biotin-tyramide) transiently. Biotinylated proteins were then isolated using Streptavidin resin for protein identification by mass spectrometry (Fig. 1b). Based on this procedure, we identified 511 potential interactors of RNF214 common to both N-terminal and C-terminal labeling (Fig. 1c). The KEGG pathway enrichment analysis revealed the Hippo pathway as the most prominent pathway to interact with RNF214 (Fig. 1d). Notably, all four human TEAD proteins, which are the final transcription factors of the Hippo pathway[36], were on the top of the list among potential interactors of RNF214 (Fig. 1d).

Next, we confirmed the interaction between RNF214 and TEAD1 using a reciprocal co-immunoprecipitation method (co-IP) after co-expressing Flag-RNF214 and Myc-TEAD1 in HEK293T cells (Fig. 1e). Similar interactions were observed between a Flag-tagged RNF214 and HA-tagged TEAD3 (Fig. 1f) or TEAD4 (Fig. 1g) or between a Myc-tagged RNF214 and Flag-TEAD2 (Supplementary Fig. 1d). In addition, Flag-tagged RNF214 was copurified with endogenous pan-TEAD and TEAD2 in HEK293A cells (Fig. 1h and Supplementary Fig. 1e) or Hep3b cells (Supplementary Fig. 1f). More significantly, endogenous RNF214 could interact with endogenous TEAD2 in Hep3b cells (Fig. 1i) and HLF cells (Supplementary Fig. 1g).

To determine the direct interactions between RNF214 and TEADs, we first purified GST-tagged TEAD1 (GST-TEAD1) recombinant proteins using a bacteria expression system (Supplementary Fig. 1h) and Strep-tagged RNF214 (Strep-RNF214) using the baculovirus-insect cell expression system (Supplementary Fig. 1i), and then performed a GST pulldown assay using these purified recombinant proteins (Fig. 1j). Indeed, GST-TEAD1 interacts with Strep-RNF214 directly. Taken together, these results established potential roles of RNF214 in the Hippo pathway.

### RNF214 enhances Hippo-regulated transcription

Since RNF214 associates with multiple downstream transcription factors of the Hippo pathway, we decided to determine whether RNF214 is involved in regulating the expression levels of genes targeted by the Hippo signaling. We first knocked down *RNF214* using small interference RNA (siRNA) in Hep3b cells and noticed that mRNA levels of three target genes, including *ANKRD1*, *CTGF*, and *CYR61*, of the TEAD transcription factors were significantly reduced in *RNF214*-knockdown cells (Fig. 2a and Supplementary Fig. 2a). Expressing an siRNA-resistant *RNF214* cDNA reinstituted mRNA levels of these three genes, especially *CTGF* and *CYR61*, excluding any potential off-target issues of siRNA

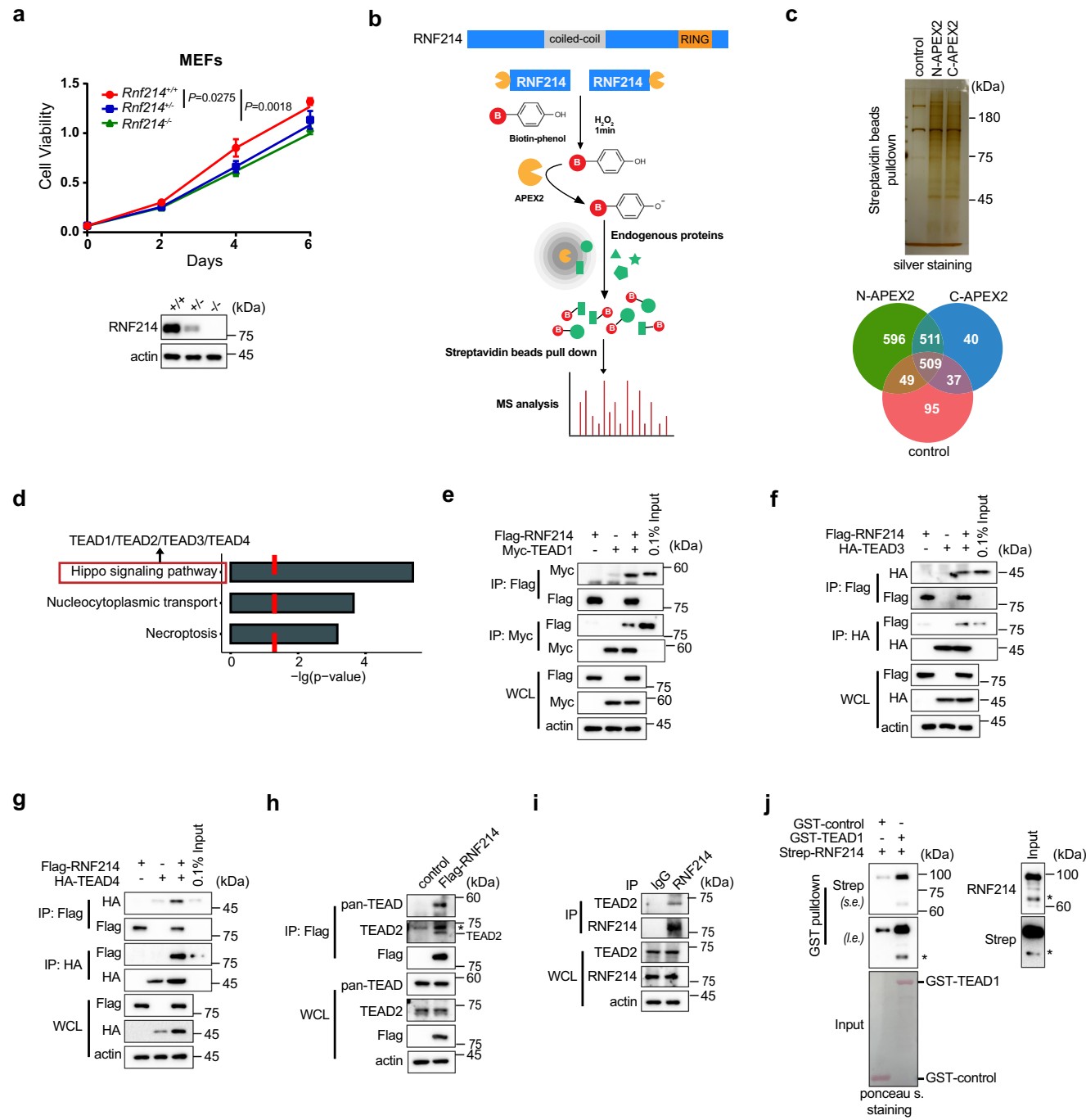

**Fig. 1 | RNF214 interacts with the TEAD transcription factors. a** Cell proliferation assay of *Rnf214* +/+, +/− and −/− MEFs. The cell viability of MEF cells was quantified by CCK8 assay. **b** Schema showing APEX2-catalyzed biotinylation. APEX2 (orange) was fused at the N-terminus or C-terminus of RNF214 (blue). Live cells were incubated with biotin-phenol and $H_2O_2$ to initiate biotinylation. APEX2 catalyzes one-electron oxidation of biotin-phenol into a biotin-phenoxyl radical, which covalently tags proximal endogenous proteins (green). Biotin-labeled proteins (red B = biotin) were enriched by Streptavidin beads and then subjected to mass spectrometry analysis. **c** Silver staining of the biotinylated proteins. The negative control with APEX2 omitted, was also treated with biotin-phenol and $H_2O_2$. Three major bands in the negative group corresponded to endogenous biotinylated proteins. Venn diagram illustrated the number of proteins identified using mass spectrometry. **d** Barplot of the KEGG pathway enrichment analysis. The Hippo pathway was significantly enriched and all four TEAD transcription factors were on the top list. *P* value was calculated through Chi-Squared test and the red dotted line means

p < 0.05. **e**–**g** RNF214 interacts with TEAD1, TEAD3 and TEAD4. HEK293T cells were transfected with Flag-RNF214 and Myc-TEAD1, HA-TEAD3 or HA-TEAD4, and reciprocal co-IP was performed using indicated antibodies in the figures. 0.1% input meant 0.1% of whole cell lysates which were used for IP. **h** Flag-RNF214 interacts with endogenous pan-TEAD and TEAD2. Flag-RNF214 was expressed in HEK293A cells which were then treated with 1 μM nocodazole for 15 min. "*" indicates non-specific band. **i** Endogenous TEAD2 interacts with RNF214. RNF214 was immuno-precipitated by home-made RNF214 antibody (J044) from Hep3b cells which were treated with 1 μM nocodazole for 15 min in advance. IgG antibody was used as the negative control. **j** TEAD1 directly interacts with RNF214 in vitro. GST was used as a negative control. "*" indicates RNF214 isoform2, which lacks 52-206 amino acids in the N-terminus. Data are presented as mean ± SD, and *P* values were calculated using two-sided unpaired Student's t-test from 3 biologically independent samples (**a**). Experiments in figures (**c**, **e**–**j**) were repeated twice. Source data are provided as a Source Data file.

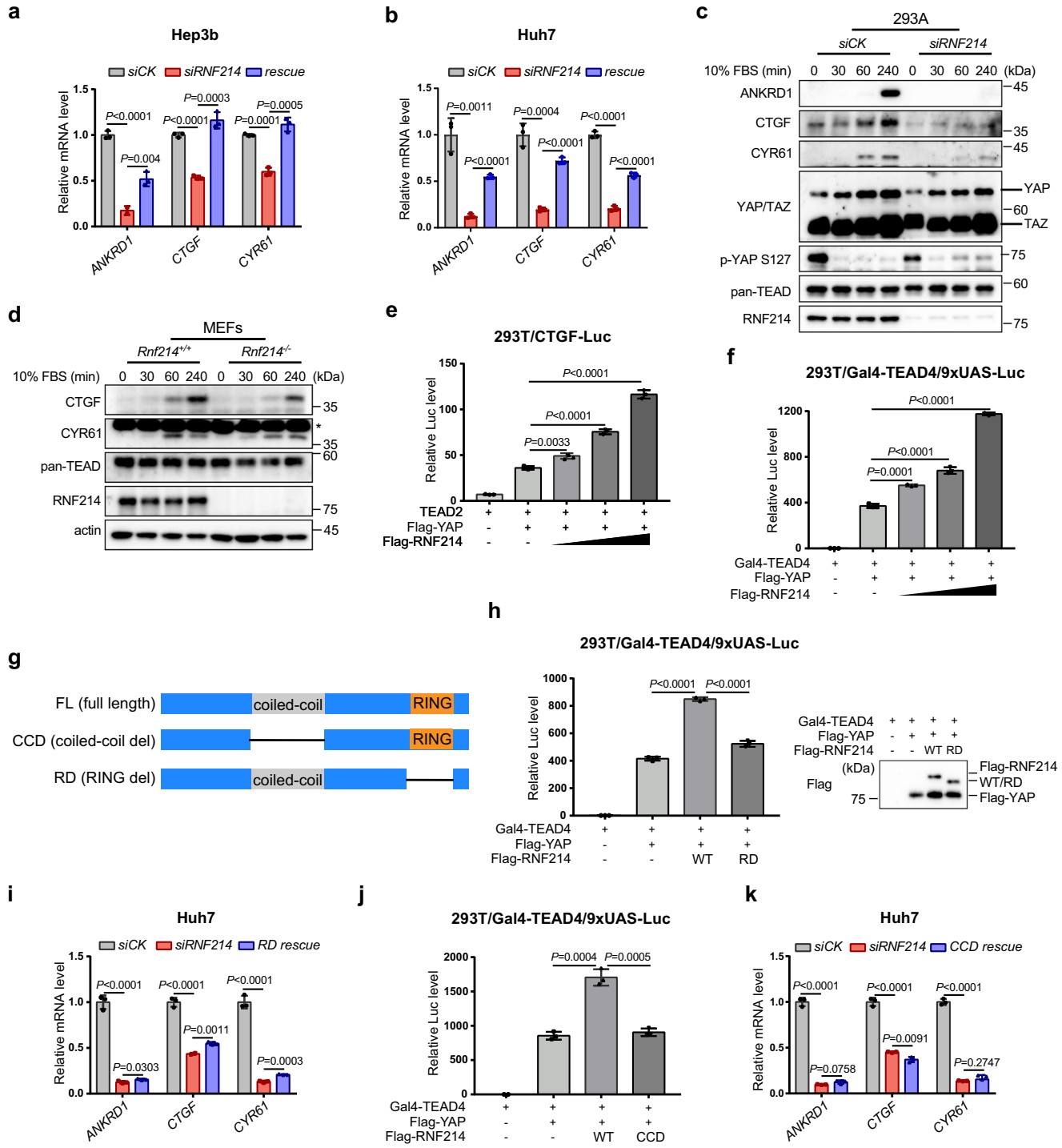

(Fig. 2a and Supplementary Fig. 2a). Of note, the *ANKRD1* mRNA level was not back to basal level after cDNA rescue of *RNF214*, suggesting there might be other regulatory mechanisms which are still poorly characterized. We also reconfirmed these results by knocking down *RNF214* in Huh7, another HCC cell line (Fig. 2b and Supplementary Fig. 2b). More significantly, we isolated two pairs of MEF cells from two litters independently and noticed that ANKRD1 and CTGF were downregulated in *Rnf214*[−/−] MEFs (Supplementary Fig. 2c).

Serum is a stimulating signal for YAP/TAZ activity and regulation of Hippo target genes[37]. Indeed, the expression of the three target genes mentioned above was blocked following serum starvation in HEK293A cells, and addition of serum resulted in their transcriptional enhancement as previously reported[37,38] (Fig. 2c). Silencing *RNF214*

inhibited the enhanced expression of these three genes observed after serum stimulation (Fig. 2c). Besides, we reconfirmed these results in *Rnf214*[−/−] MEFs (Fig. 2d), further indicating that RNF214 participates in regulating expression of Hippo target genes.

Since RNF214 is prominent for the expression of CTGF, a bona fide transcription target of the TEAD transcription factors in the Hippo pathway[36], we performed a dual luciferase assay using the *CTGF* promoter to control the expression of the firefly luciferase in HEK293T cells. While TEAD2 alone only produced a small quantity of luciferase activities, co-expressing YAP made a big increase in luciferase activities. Moreover, adding different quantities of Flag-RNF214 further enhanced luciferase activities proportionally (Fig. 2e). Consistently, Flag-RNF214 magnified both TEAD1 and TEAD3-induced

**Fig. 2 | RNF214 augments Hippo-regulated transcription. a** mRNA analysis of TEADs target genes in *RNF214*-knockdown Hep3b cells. An siRNA-resistant cDNA of *RNF214* resumed the mRNA levels of three target genes in *RNF214*-silenced cells. **b** mRNA analysis of TEADs target genes in Huh7 cells with *RNF214* knockdown. **c** Serum induces ANKRD1, CTGF and CYR61 transcription. HEK293A cells were transfected with *siRNF214*, starved in serum-free medium for 12 h and then stimulated with 10% serum for the indicated time. **d** *Rnf214*⁻/⁻ MEFs show low activity of serum induced-TEAD transcription. MEF cells were starved in serum-free medium for 12 h and then stimulated with 10% serum. "*" indicates non-specific band. **e** CTGF-luciferase reporter assay. HEK293T cells were co-transfected with the reporter system along with the increasing amounts of Flag-RNF214 (0, 100, 200, or 400 ng). **f** Gal4-TEAD4/9xUAS-luciferase reporter assay. The transcriptional activities of YAP-TEAD4 were measured based on YAP's ability to co-activate the Gal4 DNA binding domain fused to TEAD4 (Gal4-TEAD4) on the 9xUAS-luciferase reporter. Increasing amounts of Flag-RNF214 (0, 100, 200 or 400 ng) were co-transfected into the HEK293T cells with the reporter system. **g** Schematic diagram of RNF214 domains. **h, i** RNF214 enhances TEADs transcriptional activities depending on its ubiquitin ligase activity. **h** HEK293T cells were transfected with Flag-RNF214 wild type (WT) or RING finger deletion mutant (RD) along with Gal4-TEAD4/9xUAS-luciferase. Western blotting was employed to verify expression consistency between WT and RD RNF214. **i** An siRNA-resistant cDNA of *RNF214 RD* mutant was stably introduced into Huh7 cells and then endogenous *RNF214* was knocked down. **j, k** The coiled-coil domain of RNF214 is essential for its effect. **j** HEK293T cells were transfected with Flag-RNF214 WT or the coiled-coil deletion mutant (CCD) along with Gal4-TEAD4/9xUAS-luciferase. **k** An siRNA-resistant cDNA of *RNF214 CCD* mutant was stably delivered into Huh7 cells and then endogenous *RNF214* was knocked down. Data are presented as mean ± SD. *P* values were calculated using two-sided unpaired Student's *t* test; n = 3 biologically independent samples in experiments (**a, b, e, f, h**–**k**). Experiments in figures (**c, d**) were repeated twice. Source data are provided as a Source Data file.

CTGF-luciferase activities (Supplementary Fig. 2d), suggesting that RNF214 increases transcription activities of TEADs as a whole. We also employed the Gal4-TEAD4/9×UAS luciferase reporter assay[36,38]. Co-expressing Gal4-TEAD4 and YAP produced some luciferase activities, but adding Flag-RNF214 further augmented luciferase activities significantly (Fig. 2f). Of note, the enhancement of luciferase activities was proportional to the expression level of Flag-RNF214 (Fig. 2f). All together, these data demonstrated that RNF214 works together with the TEAD transcription factors to control the expression of downstream target genes of the Hippo pathway.

RNF214 is a family member of the RING finger ubiquitin ligases (Fig. 2g). Therefore, we created an RNF214 mutant with its RING finger domain deleted (RD). Unlike the wild-type RNF214, the RD mutant behaved like the transfection control in the Gal4-TEAD4/9×UAS luciferase assay when overexpressed in HEK293T cells (Fig. 2h). It also could not reinstate the expressions of *ANKRD1*, *CTGF*, and *CYR61* when introduced into *RNF214* knockdown Huh7 cells (Fig. 2i and Supplementary Fig. 2e), verifying the critical role of RNF214 as a ubiquitin ligase in the Hippo pathway. Beside the RING finger domain at its C-terminus, RNF214 contains a coiled-coil domain (Fig. 2g). We then constructed an RNF214 mutant with its coiled-coil domain removed (CCD), and showcased that this CCD mutant was incapable of rescuing the phenotypes of *siRNF214* in both luciferase assay (Fig. 2j), and expression analysis (Fig. 2k and Supplementary Fig. 2f). The coiled-coil domain is often involved in protein-protein interactions, especially self-oligomerization of proteins harboring it[39]. Thus, we speculated that the coiled-coil domain is used for RNF214's oligomerization which is usually employed as a mechanism to activate certain ubiquitin ligases[7,40,41] (Supplementary Fig. 2g). We then made both HA-tagged and Flag-tagged RNF214, and found RNF214 did self-associate with each other (Supplementary Fig. 2h). More importantly, this self-interaction depends on its coiled-coil domain (Supplementary Fig. 2i). These data might explain the reason why the CCD mutant couldn't rescue the *siRNF214* phenotypes, implying a potential mechanism by which the coiled-coil domain functions in RNF214 activation.

**RNF214 promotes nonproteolytic polyubiquitylation of TEADs**
RNF214 is a RING finger-containing ubiquitin ligase. As far as we know, no substrate has been identified for RNF214 yet. Having figured out that RNF214 interacts with the TEAD transcription factors and the RING finger domain of RNF214 is important for TEAD-regulated transcription, we decided next to determine whether RNF214 could ubiquitylate TEADs.

Firstly, we co-expressed HA-tagged ubiquitin (HA-Ub) and Flag-tagged TEAD2 or TEAD3 in HEK293T cells, and then employed anti-Flag antibody resins to immunoprecipitate Flag-TEAD2 (Fig. 3a) or Flag-TEAD3 (Fig. 3b). Anti-HA Western blotting showed both Flag-TEAD2 (Fig. 3a upper panel) and Flag-TEAD3 (Fig. 3b upper panel) were heavily ubiquitylated. Overexpressing Myc-tagged RNF214 (Myc-RNF214) did not alter expression levels of TEADs, but greatly enhanced ubiquitylation of either Flag-TEAD2 (Fig. 3a) or Flag-TEAD3 (Fig. 3b). More significantly, the Myc-RNF214 RD mutant could not increase ubiquitylation of either Flag-TEAD2 or Flag-TEAD3 proteins (Fig. 3a, b upper panel). These data confirmed that RNF214 is a ubiquitin ligase of the TEAD family proteins.

Secondly, we expressed an Avi-tagged TEAD2 in the HLF where we co-expressed BirA, a bacteria biotin ligase which conjugates biotin to the Avi-tag, a biotin-acceptor peptide. We then pulled out biotinylated Avi-TEAD2 (Bio-TEAD2) proteins using streptavidin resins under a denaturing buffer condition, and detected TEAD2 ubiquitylation using an anti-ubiquitin antibody. Clearly, Flag-RNF214 increased ubiquitylation of TEAD2 in both HLF (Fig. 3c) and Huh1 cells (Supplementary Fig. 3a). Using the same approach, we noticed that the wild-type RNF214 could augment TEAD4 ubiquitylation, while the RD mutant could not (Supplementary Fig. 3b). In addition, depletion of endogenous *RNF214* significantly decreased TEAD2 ubiquitylation in HLF cells (Fig. 3d).

Thirdly, we utilized the Halo-ThUBDs resin, a ubiquitin chain-binding matrix[42], to pull ubiquitylated proteins out of HLF cells. Using a pan-TEAD antibody, we confirmed the ubiquitylation of TEAD proteins (Fig. 3e). When we knocked out *RNF214* using the CRISPR/Cas9 method in HLF cells, TEAD ubiquitylation was largely reduced (Fig. 3e). Two independent *RNF214*-knockout clones produced similar results, indicating the observed phenotypes were not due to off-target effects from sgRNA.

One main outcome of protein ubiquitylation is proteolysis in the proteasome. When *RNF214* was either overexpressed (Fig. 1) or silenced (Fig. 2c and Supplementary Fig. 3c) or knocked out (Fig. 3e), protein levels of TEADs did not change. Besides, results from cycloheximide (CHX) chase experiments showed that TEADs were stable proteins in Hep3b cells (Fig. 3f) and HLF cells (Supplementary Fig. 3d) and *RNF214*-knockdown or knockout did not alter protein stabilities of TEADs. Moreover, TEADs were still stable when *RNF214* was overexpressed in HEK293A cells (Supplementary Fig. 3e), implying that RNF214 promotes nonproteolytic ubiquitylation of TEADs.

To determine whether RNF214 conjugates nonproteolytic poly-ubiquitin chains on TEADs, we co-expressed Flag-TEAD2, and the wild type, single lysine-only or single lysine to arginine (KR) mutants of HA-tagged ubiquitin in HEK293T cells. We observed that the K27-only ubiquitin mutant supported the basal ubiquitylation of TEAD2 as well as the wild type did (Fig. 3g), while the K27R mutant could not (Fig. 3h). Importantly, we observed that the K27-only mutant of ubiquitin supported Myc-RNF214-enhanced TEAD2 ubiquitylation (Fig. 3i), further indicating that RNF214 might mainly conjugate non-proteolytic K27 polyubiquitin chains on TEADs, although we could not rule out the possibility that RNF214 might synthesize mixed polyubiquitin chains on TEADs.

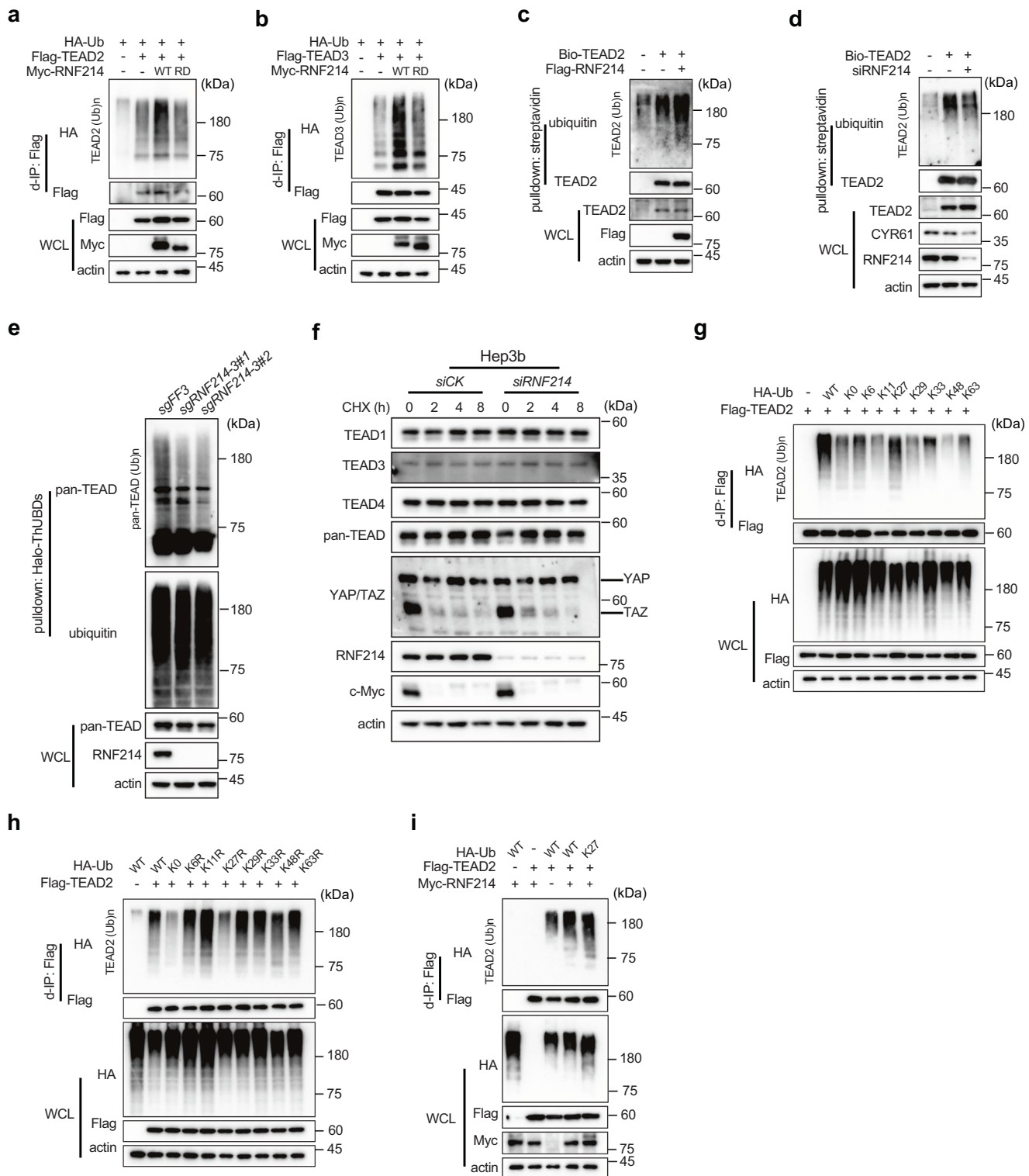

## RNF214 enhances the interactions between TEADs and YAP

The TEAD family transcription factors have little transcriptional activity by themselves and require the presence of transcription coactivators YAP or TAZ to induce target gene transcription[36,43–45]. Thus, YAP/TAZ nuclear localization and interactions with TEADs are two critical steps for the TEADs-controlled transcription. Besides, as transcription factors, TEADs' activities are also regulated by nuclear-cytoplasmic localization upon cellular stress, like many other transcription factors, such as NF-κB and SMAD[44,46]. Plus, it is also well documented that non-proteolytic polyubiquitin chains could be

signals for changes of protein subcellular localizations[47,48]. Since RNF214 conjugates nondegradable polyubiquitin chains on TEADs, we first wondered whether RNF214 could influence subcellular localizations of TEADs and YAP. In many cancer cells, YAP is always highly activated and accumulates in the nucleus[49,50]. As we observed in Huh7 cells, both TEAD1 and YAP mainly localized in the nucleus and silencing *RNF214* had no effect on subcellular localizations of TEAD1 and YAP (Supplementary Fig. 4a, b). In the HEK293A cells, YAP mainly localizes in the cytoplasm and translocated into the nucleus under Nocodazole stimulation[51]. Nocodazole disrupts microtubule polymerization and

**Fig. 3 | RNF214 promotes nonproteolytic polyubiquitylation of TEADs.**
**a, b** RNF214 promotes TEADs ubiquitylation. HEK293T cells were transfected with HA-Ub, Flag-TEAD2 or Flag-TEAD3 and Myc-RNF214 or the Myc-RNF214 RD mutant plasmids. Flag-tagged TEAD proteins were pulled out using anti-Flag beads by denaturing immunoprecipitation (d-IP) and the ubiquitylated TEAD proteins were detected using anti-HA antibody. **c** RNF214 ubiquitylates TEAD2. A biotinylated Avi-tagged TEAD2 (Bio-TEAD2) was expressed in HLF cells. Flag-RNF214 was then transfected into the Bio-TEAD2 HLF cells and biotin (2 μg/mL) was added to culture medium overnight before cell harvest. Biotinylated-TEAD2 proteins were isolated through Streptavidin beads under a denaturing buffer condition and ubiquitylated TEAD2 proteins were then detected using an anti-ubiquitin antibody. **d** Depletion of RNF214 attenuates TEAD2 ubiquitylation. Bio-TEAD2 HLF cells were transfected with *RNF214* siRNA or control siRNA. After transfection, the ubiquitylation of TEAD2 was detected using the same procedure as shown in (**c**). **e** Knockout of *RNF214* decreases ubiquitylation of TEADs. Halo-ThUBDs proteins were expressed and purified, and then incubated with HLF cell lysates. Ubiquitylated TEADs were detected using a pan-TEAD antibody. Two independent clones were selected from *sgRNF214-3* HLF pools. **f** RNF214 has little effect to the TEADs protein stability. Hep3b cells were transfected with *siRNF214* for 72 h, and then treated with cycloheximide (CHX 20 μg/mL) for the indicated time. c-Myc was employed as a positive control for CHX chase experiments. **g–i** RNF214 promotes the K27 polyubiquitylation of TEAD2. HEK293T-cells were transfected with Flag-TEAD2, and wild type, lysine less (K0), or K-only ubiquitin mutants (**g**) or KR mutants of HA-Ub (**h**). 24 h post-transfection, Flag-TEAD2 proteins were pulled out using anti-Flag beads under a denaturing buffer condition and the ubiquitylated TEAD2 proteins were reviewed using anti-HA antibody. **i** Flag-TEAD2, HA-Ub WT or the K27-only mutant with or without Myc-RNF214 were co-expressed in HEK293T cells. Experiments in these figures (**a–i**) were repeated twice. Source data are provided as a Source Data file.

induces YAP dephosphorylation and nuclear translocation[51]. The cytoplasmic-nuclear shuttling of YAP wasn't blocked when *RNF214* was silenced (Supplementary Fig. 4c). In addition, *RNF214* knockdown or overexpression had little impact on the protein levels of YAP/TAZ and non-phospho YAP (active YAP) in Hep3b or HEK293A cells (Fig. 2c, Supplementary Fig. 3c and Supplementary Fig. 4d). Together, these data suggested that RNF214 does not affect cellular localizations or cytoplasmic-nuclear shuttling of YAP and TEADs.

Next, we asked whether RNF214 could influence the interactions between YAP and TEADs. As shown in Fig. 4a, Myc-RNF214 boosted the interaction between Flag-YAP and HA-TEAD2. Interestingly, the enhancement of interaction between HA-TEAD2 and Flag-YAP largely disappeared when the RD mutant of RNF214 was employed (Fig. 4a). Similar results of the interaction between HA-TEAD4 and Flag-YAP were achieved (Fig. 4b). Consistent results were obtained between HA-TEAD1 and Flag-YAP (Supplementary Fig. 4e), further echoing the importance of the ubiquitylation activity of RNF214 in regulating the Hippo pathway. Furthermore, we found that HA-RNF214 interacts weakly with Flag-YAP in HEK293T cells (Supplementary Fig. 4f), which was disappeared in *TEAD1/3/4* knockdown HEK293T cells, establishing the important roles of RNF214 in the YAP-TEAD transcription complex (Supplementary Fig. 4g).

To further demonstrate that the ubiquitylation of TEADs by RNF214 is important for their interactions with YAP and subsequent transcriptional activities, we intended to identify the ubiquitylation sites of TEADs in an RNF214-dependent manner. Since RNF214-enhanced interactions between YAP and TEADs depend on its ubiquitin ligase activity (Fig. 4a, b and Supplementary Fig. 4e), we turned to the YAP binding domains (YBD) of TEADs. There are eight lysine residues which are conserved among the YBD domains of TEADs. Excluding those lysine residues on the YAP binding surface or those potentially affecting structural stability of TEADs[52–54], we focused on four lysine residues of TEAD2 (Fig. 4c). We made two TEAD2 mutants containing lysine-to-arginine (KR) substitutions on these lysine residues (K345R and 3KR containing K280R, K281R, and K351R), and analyzed ubiquitylation of TEAD2 mutants in HEK293T cells. In comparison with the wild type, the K345R mutant, rather than the 3KR mutant, completely lost the enhanced ubiquitylation of TEAD2 by exogenous Myc-RNF214 (Fig. 4d, e). Moreover, RNF214 failed to enhance the interaction between YAP and the K345R mutant of TEAD2 (Fig. 4f). In comparison to the wild type TEAD2, the K345R mutant failed to fully support RNF214-induced CTGF-driven luciferase activities in HEK293T cells (Fig. 4g). The K260 residue of TEAD4 in Gal4-TEAD4 corresponds to the K345 residue of TEAD2. Similarly, Flag-RNF214 couldn't rescue the luciferase activities controlled by the K260R mutant of Gal4-TEAD4 as high as by the wild type Gal4-TEAD4, in *TEAD1/3/4* knockdown HEK293T cells (Fig. 4h, i). Together, these data confirmed that the K345 residue of TEAD2 is the major ubiquitylation site mediated by RNF214 and validated the importance of RNF214 in the Hippo-mediated transcription via ubiquitylating TEADs likely at a single lysine site.

Finally, we wondered how TEAD ubiquitylation by RNF214 affects their interactions with YAP. One possibility is that YAP might possess polyubiquitin chain binding features to enhance YAP's recruitment to the TEAD transcriptional complex. To verify this hypothesis, we did a GST pulldown assay using purified recombinant proteins, including GST-YAP or GST-TAZ and synthetic polyubiquitin chains. We observed both YAP and TAZ directly bound to K48 and K63 polyubiquitin chains (Fig. 4j, k), suggesting that they are ubiquitin-binding proteins. These data might explain why TEADs ubiquitylation promotes their interactions with YAP.

All together, these data mechanistically demonstrated the importance of TEADs ubiquitylation by RNF214 in their interactions with YAP/TAZ.

## Overexpression of RNF214 correlates with poor prognosis in HCC

YAP and TEAD proteins are the key downstream effectors in the Hippo pathway and oncogenic proteins in common cancer types[13,49,55–57]. Because RNF214 ubiquitylates TEAD proteins and promotes the interactions between TEADs and YAP, we wondered whether *RNF214* is also an oncogene implicated in tumorigenesis. We first analyzed the expression profiles of *RNF214* in the cancer-based TCGA database. Interestingly, we found that the mRNA levels of *RNF214* are upregulated in HCC (Fig. 5a). Similar results were obtained from Tiger, another cancer-related database[58] (Supplementary Fig. 5a). Compared with non-tumor tissues, HCC tumor samples contain much higher mRNA levels of *RNF214* (Fig. 5a and Supplementary Fig. 5a). Further Kaplan-Meier analysis of overall survival and progression-free survival in the TCGA database showed a reverse correlation between *RNF214* expression level and the survival probability (Fig. 5b and Supplementary Fig. 5b). We then examined protein expression levels of RNF214 in a published dataset containing protein quantification of 6478 genes between 159 pairs of tumor and non-tumor samples[59], and found that protein expression levels of RNF214 are higher in tumor samples than in paracancerous ones (Fig. 5c).

To further validate the results of these statistical analysis, we compared RNF214 expression levels between 176 pairs of HCC tumor samples and paracancerous tissues from Zhejiang Provincial People's Hospital using an immunohistochemistry (IHC) approach, and noticed that RNF214 was overexpressed in tumor samples among 92 pairs, accounting for 52.3% (Fig. 5d–f), indicating that HCC tumor samples in over half of HCC patients possessed upregulated protein levels of RNF214. Meanwhile, we analyzed the correlation between RNF214 expression levels and differentiation grades of 275 cases of HCC patients based on our IHC results, and uncovered that more than half of the cases with either medium or low differentiation grades displayed high expression level (IHC score ≥6) of RNF214 with an R value

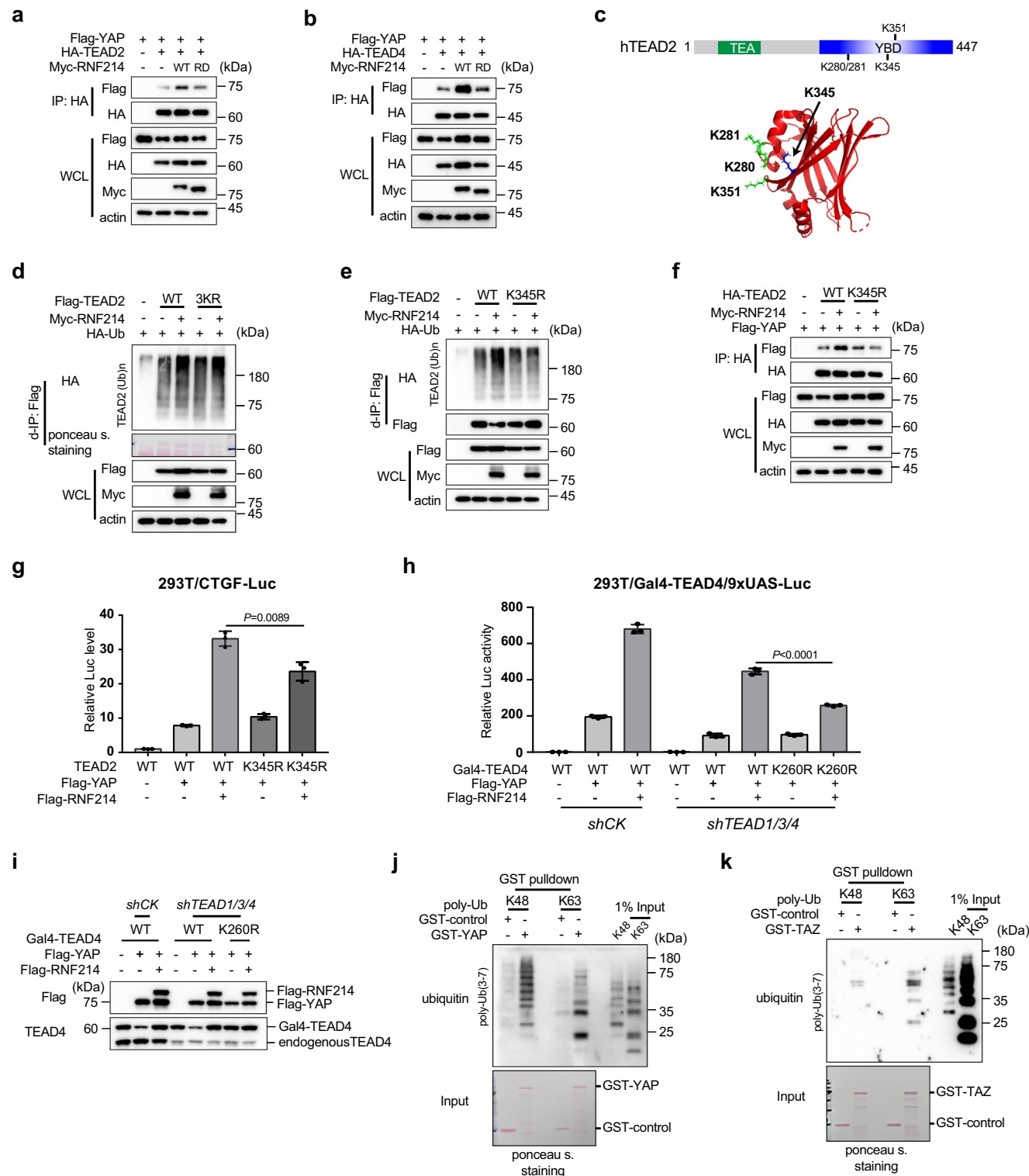

at 0.26 (Fig. 5g–i), implicating that RNF214 might contribute to the malignancy of HCC. Comparable results were acquired based on the correlation of RNF214 expression and Edmonson-Steiner grade, another crucial prognosticator in HCC (Supplementary Fig. 5c, d). In this case, 79.6% and 43.3% of patients of grade III and IV had high expression levels of RNF214 (IHC score ≥6), respectively. Moreover, we observed that RNF214 protein levels were closely associated with serum expression levels of alpha-fetoprotein (AFP), a bona fide liver cancer biomarker (Supplementary Fig. 5e). To further consolidate our

clinical analysis, we measured RNF214 expression in HCC cell lines using a Western blotting approach and noted that RNF214 protein levels were elevated in all seven HCC cell lines examined compared with HL7702, a normal liver cell line (Fig. 5j).

We then examined the relationships between *RNF214* and Hippo-regulated gene expression in liver cancer samples. Through Spearman's rank correlation analysis of the TCGA cohort, we observed a positive correlation between *RNF214* and YAP/TAZ-TEAD target genes (e.g., *AMOTL2, CTGF, CYR61, ANKRD1, AXL, BCL2, CCND1,* and *CDH2*)

**Fig. 4 | RNF214 enhances the interactions between TEADs and YAP. a, b** RNF214 increases the interactions between YAP and TEADs. HEK293T cells were transfected with YAP, TEAD and RNF214 WT or RD mutant. co-IP and immunoblotting were performed as indicated in the figure. **c** Schematic diagram of 4 lysine (K) residues potentially ubiquitylated in TEAD2 YBD domain. Transcriptional enhanced associate (TEA); YAP binding domain (YBD). RNF214 ubiquitylates TEAD2 on K345 residue. HEK293T cells were transfected with HA-Ub, Myc-RNF214, Flag-TEAD2 wild type (WT) and 3KR mutant (**d**) or the K345R mutant (**e**). Flag-TEAD2 proteins were immunoprecipitated using anti-Flag beads and the ubiquitylated TEAD2 proteins were detected using anti-HA antibody. TEAD2 3KR mutant contains KR substitutions on K280, K281 and K351 residues. **f** RNF214 fails to promote the interaction between YAP and the K345R mutant of TEAD2. HEK293T cells were transfected with Myc-RNF214, Flag-YAP and HA-TEAD2 WT or the K345R mutant. **g** The TEAD2

K345R mutant fails to fully support RNF214-induced CTGF-driven luciferase activities in HEK293T cells. **h, i** The Gal4-TEAD4 K260R mutant fails to rescue the RNF214-induced luciferase activities in *TEAD1/3/4* knockdown cells. The K260 residue of TEAD4 in Gal4-TEAD4 corresponds to the K345 residue of TEAD2. Expression consistency was verified by Western blotting. **j, k** YAP and TAZ directly bind to K48 and K63 polyubiquitin chains in vitro. Pulldown assays to determine whether YAP or TAZ possess polyubiquitin chain binding abilities. GST-YAP or GST-TAZ were purified from BL21 (DE3) bacteria cells and poly-K48 Ub (3-7) and poly-K63 Ub (3-7) chains were purchased from R&D Systems. GST was used as a negative control. Data are presented as mean ± SD, and *P* values were calculated using two-sided unpaired Student's *t* test from 3 biologically independent samples (**g, h**). Experiments in figures (**a, b, d–f, i–k**) were repeated twice. Source data are provided as a Source Data file.

(Fig. 5k–m and Supplementary Fig. 5f–j). Besides, *RNF214* expression positively correlated with *YAP* and *TAZ* expression, as well as *TEAD1-4* expression (Supplementary Fig. 5k, l).

Together, these data suggested that *RNF214* could be a critical oncogene and tightly associated with enhanced YAP/TAZ-TEAD transcription activities in promoting HCC tumorigenesis.

### RNF214 is critical for HCC tumorigenesis

To investigate the functions of RNF214 in HCC, we first knocked out *RNF214* in HLF cells, using the CRISPR/Cas9 method and found that all of the small guide RNAs (sgRNAs) slowed the growth of HLF cells (Fig. 6a). These *RNF214* knockout HLF cells produced a smaller number of colonies than control cells in a colony formation assay (Fig. 6b upper panel). Quantitative analysis demonstrated that the differences between the *RNF214* knockout and control cells were statistically significant (Fig. 6b bottom panel). Similar phenotype was detected in Huh7 cells as well when *RNF214* was knocked out (Supplementary Fig. 6a). Meanwhile, we silenced *RNF214* in Hep3b cells, using a small hairpin RNA (shRNA) method and demonstrated that *RNF214*-silenced Hep3b cells produced fewer colonies than control shRNA cells (Supplementary Fig. 6b). Moreover, we knocked down *RNF214* in Hep3b cells using siRNA oligos and noted that *RNF214*-silenced cells propagated slower than control cells (Fig. 6c). Importantly, an siRNA-resistant cDNA of *RNF214* could resume the proliferation rate of Hep3b cells to a large extent (Fig. 6c), indicating the authenticity of these phenotypes. Conversely, we overexpressed *RNF214* in Huh1, an HCC cell line with relatively low expression of RNF214 (Fig. 5j) and observed that the number of colonies was at least doubled under this condition (Supplementary Fig. 6c). Furthermore, we found that RNF214 could promote proliferation of Hep3b cells when overexpressed (Supplementary Fig. 6d). Together, these results evidenced that RNF214 is a positive regulator of HCC cell proliferation.

To study the roles of RNF214 in migration and invasion of HCC cells, we first examined the migration ability of Hep3b cells in a wound-healing assay and found that *RNF214* knockdown cells migrated slower than control Hep3b cells (Supplementary Fig. 6e). We also performed the transwell migration assay (without Matrigel) and the transwell invasion assay (with Matrigel), respectively. Control Hep3b cells possess excellent abilities of migration and invasion, while knocking down *RNF214* using an siRNA oligo reduced abilities of migration, especially of invasion remarkably (Fig. 6d). An siRNA-resistant cDNA of *RNF214* rescued these phenotypes, implicating that these phenotypes were authentic (Fig. 6d). Altogether, these data further indicated that *RNF214* is an oncogene in HCC.

Both YAP and TEADs play eminent roles in cancer development, progression and metastasis, including HCC tumorigenesis[43,55,60–67]. Having found that RNF214 functions as a positive regulator of YAP/TAZ-TEAD transcriptional complex and promotes tumor cell properties, we decided to determine whether RNF214 is critical for the oncogenic activities of YAP and TEADs in HCC. Phosphorylation of YAP at serine-127 results in its cytoplasmic retention, whereas the non-

phosphorylatable S127A mutant becomes constitutively active in the nucleus[26]. We first created both HLF and Huh7 cell lines expressing the S127A YAP mutant using the tetracycline-inducible (Tet/on) gene expression system, and then knocked down *RNF214* using siRNA. Indeed, the S127A mutant induced higher expression levels of these three Hippo target genes in both HLF and Huh7 cells (Supplementary Fig. 6f, g). More consistently, *RNF214* knockdown could decrease mRNA expression levels of three Hippo target genes at both basal and YAP S127A-induced levels (Supplementary Fig. 6f, g). Overexpressing the S127A mutant of YAP strengthened the migration of Huh7 cells profoundly (Supplementary Fig. 6h). However, silencing *RNF214* significantly inhibited cell migration under both basal and overexpressed conditions of YAP (Supplementary Fig. 6h).

To further evaluate the roles of RNF214 in HCC tumorigenesis, we employed a subcutaneous xenograft mouse model. We subcutaneously injected $2 \times 10^6$ Huh7 cells with Matrigel into 5-week-old male BALB/c nude mice. Overall, *RNF214* knockdown Huh7 cells grew into tumors much slower than the shRNA control cells in nude mice (Supplementary Fig. 7a). Tumors grew from *RNF214*-silenced Huh7 cells were much smaller than those from shRNA control cells (Fig. 6e and Supplementary Fig. 7b, c). More relevantly, *RNF214*-silenced tumors expressed tremendously lower amount of *CYR61*'s mRNAs and proteins than control tumors (Supplementary Fig. 7d, e). Furthermore, *RNF214* knockout Huh7 cells produced smaller tumors than control cells when subcutaneously injected into nude mice (Supplementary Fig. 7f–h).

It has been reported that overexpression of both *RAS and CTNNB1* was able to drive tumorigenesis in liver cancer and tightly related to the Hippo pathway[68]. Therefore, we employed hydrodynamic tail vein injection to induce high expression of both *NRAS and CTNNB*1 in mice to further demonstrate the function of RNF214 in orthotopic liver cancer. Three plasmids were employed in a Sleeping Beauty system to co-express *NRAS and CTNNB1* together with two *Rnf214* shRNAs (Fig. 6f). Mice were sacrificed 120 days after injection and the livers were weighted and imaged (Fig. 6g and Supplementary Fig. 7i). Overexpression of both *NRAS* and *CTNNB1* promoted tumorigenesis in livers of mice, and depletion of *Rnf214* suppressed tumor formation. Consistent with these observations, the liver-to-body weight ratio was markedly decreased when *Rnf214* was knocked down (Fig. 6h). More importantly, the expressions of CTGF and CYR61 were lower in *Rnf214* knockdown tumor sections than control tumors (Fig. 6i). Finally, we applied hematoxylin-eosin staining and IHC approaches to verify the formation of tumors in livers of mice and the expression of HA-tagged NRAS, CTNNB1, and RNF214 in tumors respectively (Fig. 6j).

Together, our data concluded that RNF214 promotes HCC development and progression via governing the downstream effect of the Hippo pathway and is a bona fide oncogene in HCC (Fig. 7).

## Discussion

Protein ubiquitylation is pivotal for many essential cellular activities. Components of the ubiquitin signaling pathway have been implicated

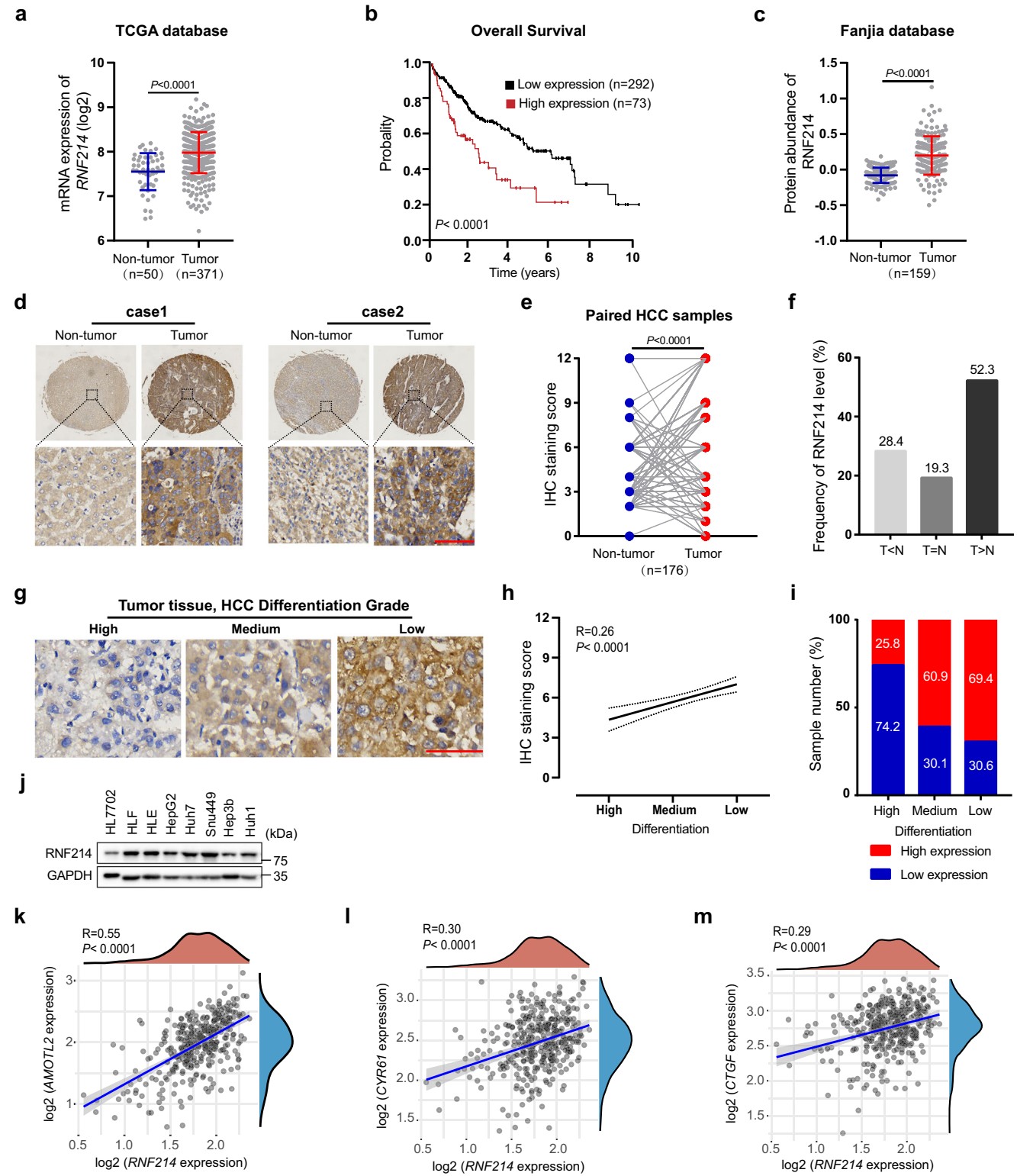

in tumor initiation, progression and metastasis in both positive and negative ways. Protein ubiquitylation is also a specific process and the specificity is mainly maintained by ubiquitin ligases which contain either a RING finger or a HECT domain[7]. RNF214 belongs to the family of the RING finger ubiquitin ligases, but its function is understudied except as a candidate gene potentially in milk lactose regulation based on a GWAS study[69]. By combining an APEX2 proximity labeling method and mass spectrometry, we identified the TEAD family proteins, major transcription factors of the Hippo pathway, as main interactors of

RNF214. Human genome encodes four TEAD proteins and all of them emerged in our mass spectrometry analysis. Of note, Barroso-Gomila et al. employed a different approach to identify potential substrates of ubiquitin ligases including RNF214[70]. In their study, they identified some potential substrates of RNF214 in HEK293FT cells, but did not find TEADs as substrates of RNF214, since we employed HLF, an HCC cell line to identify substrates of RNF214. Using a series of biochemical approaches, we validated the interactions of RNF214 with the TEAD family proteins and provided strong evidences supporting RNF214 as

**Fig. 5 | Overexpression of RNF214 correlates with poor prognosis in HCC.**
**a** Bioinformatic analysis of *RNF214* mRNA levels in HCC. *RNF214* mRNA levels from TCGA database were analyzed using two-sided unpaired Student's *t* test. n = 50 in the non-tumor group; n = 371 in the tumor group. Data are presented as mean ± SD.
**b** Kaplan-Meier survival curves of overall survival based on *RNF214* expression in TCGA database. The image was prepared using the Human Protein Atlas.
**c**, Bioinformatic analysis of RNF214 protein abundance in HCC. Protein expression levels were from Fanjia database and *P* value was analyzed through two-sided paired Student's *t* test from 159 paired tissues. Data are presented as mean ± SD.
**d–f** Immunohistochemical (IHC) staining of RNF214 in an HCC tissue microarray. Representative images were presented in (**d**). Scale bar, 100 µm. The IHC scores between paired tumor and non-tumor tissues from 176 patients were followed by two-sided paired Student's *t* test in (**e**). RNF214 protein levels are higher in cancer samples than in their paired adjacent normal tissues (**f**); T: tumors; N: paired-adjacent normal tissues. **g–i** Spearman's correlation analysis between RNF214 expression and differentiation grades in tumor tissues from 275 patients with HCC. Representative images were shown in (**g**). Scale bar, 100 µm. RNF214 (high), IHC score ≥6; RNF214 (low), IHC score <6. High differentiation (RNF214 high, n = 8; RNF214 low, n = 23); Medium differentiation (RNF214 high, n = 81; RNF214 low, n = 52); Low differentiation (RNF214 high, n = 77; RNF214 low, n = 34). **j** Western blotting of RNF214 protein expression in HCC cell lines. Experiments were repeated twice. **k–m** Expression levels of *RNF214* are positively correlated with YAP/TAZ-TEAD target genes (e.g., *AMOTL2*, *CTGF*, and *CYR61*) in liver cancer patients. The correlation of two genes was described through Spearman's correlation analysis based on TCGA dataset (n = 371).

the ubiquitin ligase of the TEAD proteins. We have observed the direct interactions between RNF214 and TEADs, however, we still don't know how RNF214 interacts with TEADs. As our data indicated, of three domains of RNF214, the RING finger domain is required for its ligase activities and the coiled-coil domain is essential for its self-association which is important for activation of RNF214. One possibility is that RNF214 recognizes TEADs via the N-terminal part of RNF214. Further study is needed to validate this hypothesis.

As transcription factors, TEADs orchestrate transcription of genes related to development, cell growth, organ size control, and onco-genesis together with YAP/TAZ, two transcriptional coactivators and major downstream effectors of the Hippo pathway[36,43,44]. Posttranslational modifications have been shown to regulate functions of TEAD proteins. For example, phosphorylation of TEAD1 by either protein kinase C or protein kinase A can significantly reduce DNA binding activity of TEAD1[24,25], whereas, palmitoylation of TEADs is crucial for their proper folding and protein stability maintenance[71–73]. Our data indicated that ubiquitylation of TEADs by RNF214 is important for their functions as downstream transcription factors of the Hippo pathway. We found that the interactions between TEADs and YAP are pro-foundly increased by the existence of the RNF214 ubiquitin ligase. More significantly, the ubiquitylation activity of RNF214 is key for their enhanced interactions, and subsequent transcription of YAP-TEAD-regulated genes. Although the ubiquitin signaling pathway has been linked to the Hippo pathway by regulating protein stabilities and localizations of several key components in the Hippo signaling, such as YAP/TAZ, LATS1/2, MOB1, and MST1/2[8,18], there were no previous studies to make any connection between ubiquitylation and the biological activities of TEADs. Our data also demonstrated that RNF214 mainly conjugates non-proteolytic polyubiquitin chains most likely on a single lysine site of TEADs. Since ubiquitylation of TEADs is important for their interactions with YAP, we speculated that YAP or additional YAP-associated proteins might possess polyubiquitin chain binding features to enhance YAP recruitment to the TEAD transcriptional complex. Indeed, our results of in vitro GST pulldown assays indicated that YAP and TAZ possess ubiquitin-binding abilities and suggested that a conserved domain between YAP and TAZ may act as a polyubiquitin-binding domain. Further studies are needed to answer this question.

The Hippo pathway has been implicated in tumorigenesis, with *MST1/2* and *LATS1/2* kinases as tumor suppressors, but *YAP/TAZ* and *TEADs* as oncogenes[9,17,74]. YAP and TEADs have been proposed as promising therapeutic targets in cancer therapy[75–78]. Indeed, small molecule inhibitors disrupting the interactions between TEADs and YAP are under development as cancer drugs[38,79–82]. By combining clinical data and biological analysis, we proved that *RNF214* is an oncogene of HCC and an important regulator of the YAP-TEAD transcription complex in general. Therefore, adding RNF214 to the axis of YAP-TEAD could offer a promising angle to invent unique therapeutic tools to kill cancer cells, especially HCC ones by managing transcriptional activities of the TEAD and YAP/TAZ complex.

## Methods

This study complies with all relevant biosafety, animal procedures, and ethical regulations. Human ethics was approved by the Medical Ethics Committee of Zhejiang Provincial People's Hospital (QT2022058). Animal protocols used in the study were approved by the Animal Ethics Committee of Zhejiang University (ZJU20240073).

### Cell culture and transfection

HEK293T (CRL-11268) cells were from ATCC. HEK293A (ATCC, CRL-1573), Snu449 (ATCC, CRL-2234), and HL7702 (Chinese Academy of Sciences Cell bank, GNHu6) cells were from Dr. Bin Zhao. HepG2 (ATCC, HB-8055), Hep3b (ATCC, HB-8064), Huh7 (JCRB cell bank, JCRB 0403), Huh1 (JCRB cell bank, JCRB 0199), HLF (JCRB cell bank, JCRB 0405), and HLE (JCRB cell bank, JCRB 0404) cells were from Dr. Jun-fang Ji. Primary MEF cells were isolated from 13.5 days' mouse embryos and the sex is not under consideration. HEK293T, HEK293A, HLF, HLE, HepG2, Huh7, Hep3b, Huh1, and MEF cells were maintained in DMEM supplemented with 10% fetal bovine serum. HL7702 and Snu449 were cultured in RPMI1640 supplemented with 10% fetal bovine serum. All cells were incubated at 37 °C, with 5% $CO_2$. Plasmids were transfected into cells using Lipofectamine 3000 (Invitrogen) according to the manufacturer's protocol.

### Plasmids

Human RNF214 coding sequence was amplified using Polymerase Chain Reaction (PCR) from a human cDNA library made in Jianping Jin's laboratory, then subcloned into a Gateway entry plasmid pENTR-1W and validated by sequencing. Truncated mutants of RNF214 were made by PCR-based mutagenesis and confirmed by sequencing. Expression plasmids for CTGF-Luciferase, Gal4-TEAD4, 9xUAS-Luciferase, CMV-β-galactosidase, Flag-YAP, Flag-YAP-S127A, Myc-TEAD1, and Myc-TEAD2 were generously provided by Bin Zhao's laboratory, and some of them were subcloned into pENTR-1W. pENTR-TEAD3 plasmid was from the Invitrogen ORF Clones library at the core facility of Life Sciences Institute, Zhejiang University. Entry clones were shuttled into different destination vectors through LR reaction (Gateway LR Clonase II Enzyme Mix, Invitrogen). pLenti-CRISPR-puro vector was used to construct sgRNA plasmids. The gRNA sequences against *RNF214* were provided in Supplementary Table 1. pLKO-ccdB-puro vector was employed to make shRNA plasmids. The shRNA sequences against *RNF214* were provided in Supplementary Table 1.

### Analysis of RNF214 in HCC Microarray

Human HCC tissues (n = 275) and adjacent non-tumor tissues (n = 256) microarray chips were created in the Department of Pathology, Zhejiang Provincial People's Hospital. 176 cases of the tissue microarray were paired samples. All cases of HCC tissues and non-tumor tissues were diagnosed clinically and pathologically. All samples were received from the patients who underwent surgical resection and signed informed consent before their operations. The protocol was

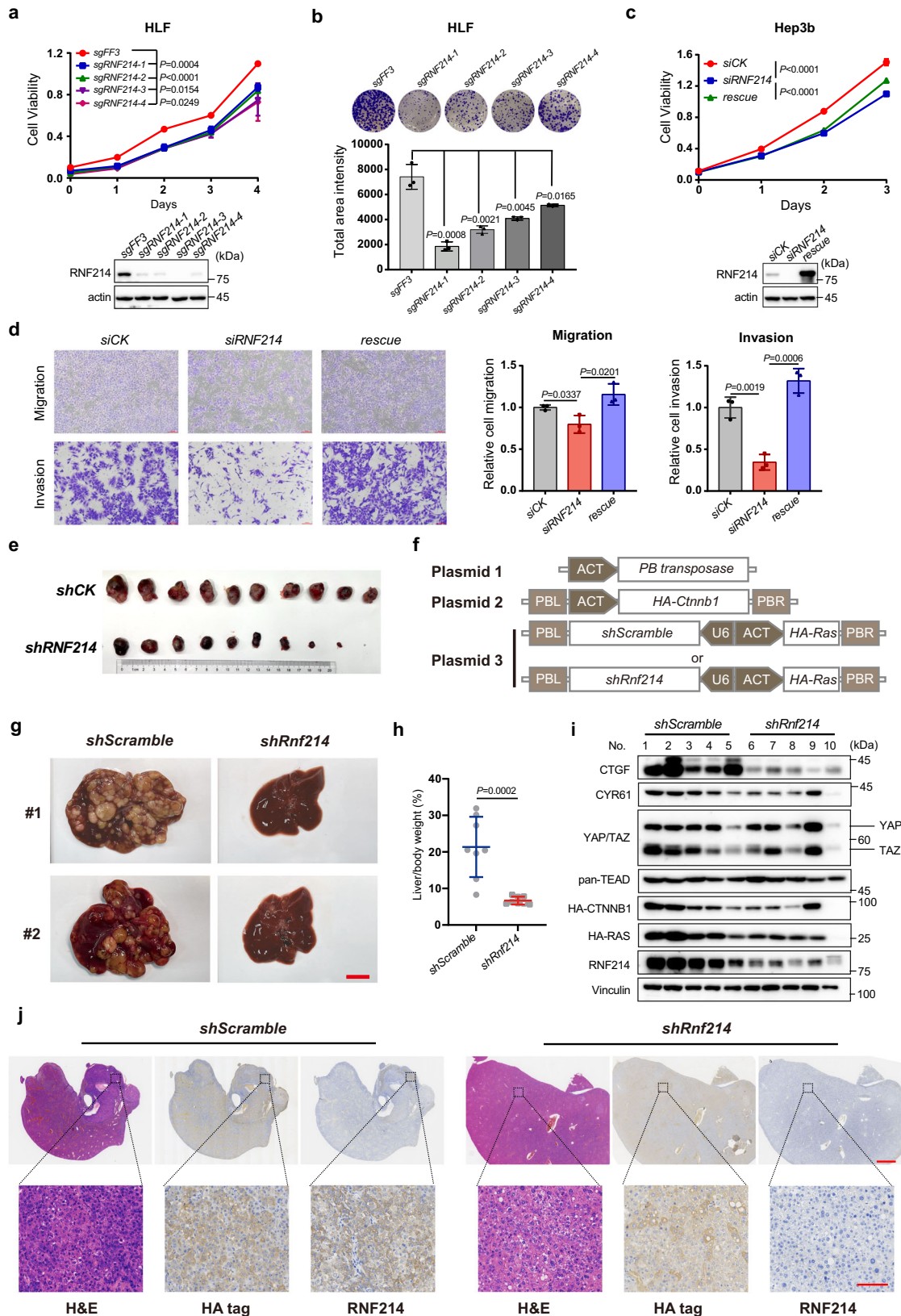

approved by the Medical Ethics Committee of Zhejiang Provincial People's Hospital.

Immunohistochemistry (IHC) was performed to detect protein levels of RNF214 on HCC microarray chips. The degree of immunostaining was reviewed and scored independently by two pathologists based on staining intensity and extent. Staining intensity was classified as 0 (negative), 1 (weak), 2 (moderate) and 3 (strong). Staining extent was divided into 0 (<5%), 1 (5–25%), 2 (26–50%), 3 (51–75%) and 4 (>75%) depending on the percentage of positive cells. IHC Score = staining intensity × staining extent.

**Fig. 6 | RNF214 is critical for HCC tumorigenesis. a, b** Proliferation in *RNF214* knockout HLF cells. **a** The cell viability of *RNF214* knockout HLF cells were quantified by CCK8 assay. n = 3 biologically independent samples. **b** Colony formation assays in *RNF214* knockout HLF cells. $10^3$ viable cells were seeded into six-well plate and incubated for 9 days. n = 3 biologically independent samples. **c** Rescue experiments of cell viability in Hep3b cells. Hep3b cells were transfected with *siRNF214* and an siRNA-resistant cDNA of *RNF214* was stably expressed in Hep3b cells using a lentivirus infection approach to rescue the growth inhibitory effect. n = 5 biologically independent samples. **d** Transwell assays of migration and invasion. Hep3b or *siRNF214*-resistant cells were transfected with siRNAs for 48 h and then plated in transwell chambers (with or without Matrigel) for another 48 h. Scale bar, 100 μm. n = 3 biologically independent samples. **e** *RNF214* knockdown inhibits HCC tumor growth in subcutaneous xenograft model. Control or *RNF214*-silenced

Huh7 cells with Matrigel were injected subcutaneously into 5-week-old male BALB/c nude mice. 21 days after cell implantation, tumors were dissected, photographed and weighted. n = 10 mice per group. **f** Schema showing the plasmids used for hydrodynamic tail vein injection mouse model. **g, h** Depletion of *Rnf214* suppressed tumor formation. Representative livers at 120 days after injection were shown (**g**). Scale bar, 1 cm. The liver-to-body weight ratio was quantified in (**h**). n = 8 mice per group. **i** Protein levels of CTGF and CYR61 in *Rnf214*-depletion tumors. The dissected tumors were subjected to Western blotting. The No. 10 tumor sample in the *shRnf214* group was very small, as a result, it is hard to detect HA-CTNNB1 and HA-RAS. **j** Hematoxylin-eosin staining (H&E), HA-tagged and RNF214 staining of mouse livers. Scale bar, 2 mm and 100 μm in scanned and zoom in figures. Data are presented as mean ± SD. *P* values were calculated using two-sided unpaired Student's *t* test from independent samples. Source data are provided as a Source Data file.

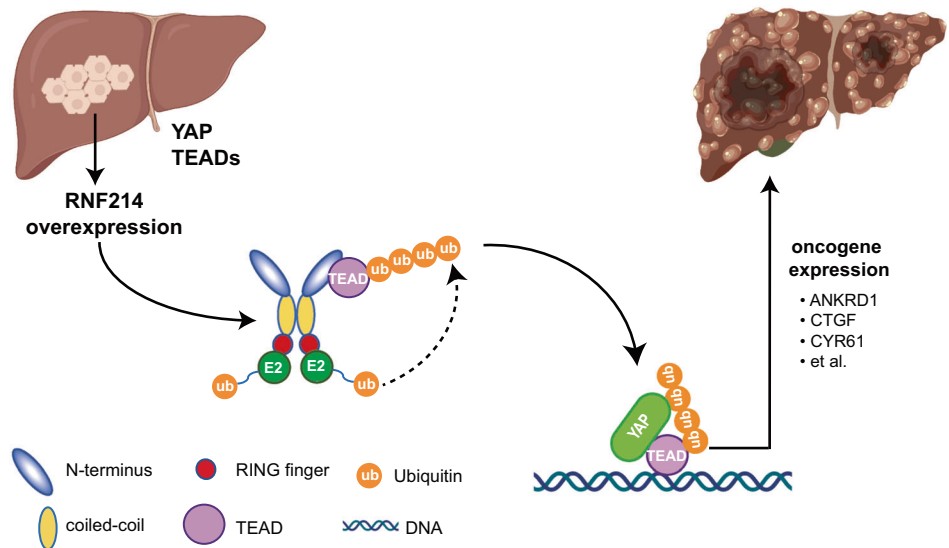

**Fig. 7 | Working model of RNF214 functions in HCC.** YAP and TAZ are transcription co-activators that activate transcriptional activities of TEADs to promote HCC tumorigenesis. RNF214 induces TEADs non-proteolytic ubiquitylation at a single conserved lysine site, enhances the interactions between TEADs and YAP,

and then promotes transactivation of the downstream genes, thereby leading to enhanced tumor progression. The figure created with BioRender. com released under a Creative Commons Attribution-NonCommercial-NoDerivs 4.0 International license (https://creativecommons.org/licenses/by-nc-nd/4.0/deed.en).

## Small interference RNA

siRNAs were transfected into HCC cells using Lipofectamine RNAiMAX Reagent (Invitrogen) according to the manufacturer's protocol. After 96 h, cells were harvested. The siRNA sequences were provided in Supplementary Table 1.

## Lentivirus production and stable cell line generation

Lentiviruses were produced by transfecting lentiviral vectors carrying target gene sequences together with the packing plasmids of psPAX2 and pMD2G into HEK293T cells using PEI. After 48 h, supernatants containing lentivirus particles were collected to infect host cells using a spin infection method. Stable cells were selected in the presence of puromycin (Sangon Biotech).

## Proliferation and colony formation assays

For proliferation assay, the viability of HCC cells was quantified by Cell Counting Kit-8 (CCK8, K1018, APExBIO). Cells with indicated treatments were seeded into 96-well plates, and incubated for the corresponding days and after 2 h of incubation with CCK8 reagents at 37 °C, absorbance at 450 nm were recorded using a microplate reader (TECAN). For colony formation assay, 6-well plates were seeded with $10^3$ viable cells and incubated for the days as indicated. At the end of the experiments, the colonies were fixed in methanol and then stained with 0.1% crystal violet. The colonies with >50 cells were counted under the microscope.

## Cell migration and invasion assays

For wound-healing assays, cells were seeded in six-well plates, grown to 100% confluence in a monolayer and then starved in serum-free DMEM overnight. After a scratch was made with a sterile pipette tip, the cells were washed with PBS and sequentially fed with serum-free DMEM. Images were acquired immediately following the "wounds" were made, and every 12 h via a microscope at 4× magnifications.

Transwell chambers (Corning) with and without precoated Matrigel were used to determine cell migration and invasion, respectively. Briefly, $6 \times 10^4$ cells in 300 μl serum-free DMEM were plated in transwell inserts and then 500 μl culture medium containing 10% FBS was added to the lower chamber. After 48 h, the cells in the upper chamber of the transwell were removed with a cotton swab, the migrated cells were fixed in methanol and stained with 0.1% crystal violet. Cells in three randomly selected fields were photographed and statistically analyzed.

## Luciferase reporter assay

For the CTGF luciferase assay, HEK293T cells were transfected with CTGF-Luciferase plasmid containing a firefly luciferase under the control of *CTGF* promoter, a Renilla luciferase plasmid as a transfection control and indicated gene expression plasmids. All values were normalized for transfection efficiency against Renilla luciferase activities. The other reporter assay was carried by transfection HEK293T cells with Gal4-TEAD4, 9xUAS-Luciferase, CMV-β-gal, and indicated

plasmids. Luciferase activities were normalized to β-gal activities. 24 h after transfection, cells were lysed and luciferase activities were measured using the Dual-Luciferase Reporter Assay System (Vazyme).

## RNA extraction and qRT-PCR

Total RNAs were isolated using Trizol reagent (Sangon Biotech). cDNAs were prepared using HiScript III 1st Strand cDNA Synthesis Kit (+gDNA wiper) (Vazyme) according to the manufacturer's protocol. The qRT-PCR analysis was performed by the SYBR green method (YEASEN). The sequences of the PCR primers for the corresponding human gene were provided in Supplementary Table 2.

## APEX2-catalyzed biotinylation and Mass spectrometry analysis

APEX2 was fused at either the N- or C-terminus of RNF214. Fusion proteins were expressed in HLF cells using a lentivirus infection method and expressed at levels comparable to the endogenous RNF214 proteins. Cells were then incubated with 2 mM biotin-phenol (APExBIO) in the DMEM supplemented with 10% fetal bovine serum for 30 min at 37 °C. Consequently, 1-min pulse with 0.25 mM $H_2O_2$ at room temperature was stopped with ice-cold quenching buffer (5 mM Trolox [Sigma], 10 mM sodium ascorbate [Sigma], and 10 mM sodium azide in PBS). All samples were washed three times with quenching buffer and then harvested.

Cell pellets were lysed in 6 M urea buffer (6 M urea, 100 mM Tris-HCl [pH 7.5], 200 mM NaCl and 1% SDS). After a short sonication, lysates were clarified by centrifugation at $21130 \times g$ and quantified using the BCA kit. Streptavidin beads (Smart-lifesciences) were washed with lysis buffer. 3 mg of each sample was mixed with 10 μL Streptavidin beads. The suspensions were gently rotated at 25 °C overnight. The beads were then washed with 6 M urea buffer five times and bound biotinylated proteins were subjected to mass spectrometry analysis. Briefly, for the reduction/alkylation reactions on beads, 200 μL 25 mM ammonium bicarbonate with 5 mM DTT (dithiothreitol) was added into the washed beads for 30 min at 56 °C, then 10 mM IAA (iodoacetamide) was added into the solution for 25 min at 25 °C. After reduction/alkylation, samples were precipitated by adding 600 μL methanol, 150 μL chloroform and 400 μL ddH$_2$O. After centrifugation at $13,523 \times g$ for 10 mins, keep the white middle layer (protein precipitation) and add 0.5 μg trypsin (Promega) into each sample solution at 37 °C for 12–16 h. After trypsin digestion and centrifugation, the supernatant samples were separated for lyophilization, and desalted by Ziptip C18 (Millipore), and then lyophilized. Then 10 μL FA (formic acid, Sigma) was added into the lyophilized and desalted peptide samples. The samples were ready to loaded into TimsTOF Pro (Bruker). 200 ng peptide samples were loaded into LC-MS system. The LC parameters were 25 cm length and 75 μm the inside diameter of LC column (IonOpticks) and the inside filler was 1.6 μm C18. The temperature setting was 50 °C. The speed was 300 nL/min and the total time was 60 min. The MS scanning range parameter was from 100 to 1700 m/z. The data was analyzed by PEAKS® Online 11 software. The parameters of database searching were as follows: Precursor Mass Error Tolerance is 15 ppm; Fragment Mass Error Tolerance is 0.05 Da; Enzyme is Trypsin; Digest Mode is Semi-Specific; Missed Cleavage is 3; Target Database is whole human proteins information from Uniprot; Peptide Length is from 6 to 45; Fixed Modification is Carbamidomethylation; Variable Modifications is Oxidation(M); Peptide-spectrum match (PSM) and Protein Group false discovery rate (FDR) are 1%.

To reveal the biological pathways of 511 proteins unique to samples from both fusion proteins, KEGG pathway enrichment analysis was performed using "clusterProfiler" R package.

## Western blotting and immunoprecipitation

Cells were lysed in lysis buffer (1%SDS and 30 μM Tris-HCl [pH6.8]). Total proteins (10 μg) were separated on SDS-PAGE and then transferred onto PVDF membranes (Millipore). After blocking using 5% nonfat milk, membranes were incubated with the gene-specific

primary antibodies, then HRP-conjugated secondary antibody (Jackson ImmunoResearch), and visualized using ECL reagents (YEASEN). Antibodies used in this study were listed in Supplementary Table 3.

For co-immunoprecipitation (co-IP), 24 h after transfection, cell lysates were lysed in 1% Triton lysis buffer (50 mM Tris-HCl [pH 7.5], 1 mM EDTA, 150 mM NaCl and 1% Triton X-100) containing protease and phosphatase inhibitors. The lysates were subjected to co-IP using specific antibody-conjugated agarose (Sigma) for 2 h. After extensive washes, immunoprecipitated proteins were separated on SDS-PAGE, transferred to PVDF membranes and detected by Western blotting with appropriate antibodies.

For endogenous and semi-endogenous immunoprecipitation, cell lysates were lysed in 1% Triton lysis buffer and sonicated for a short time. Immunoprecipitation was carried out with Flag-conjugated agarose or anti-RNF214 (J044) antibody and protein A Sepharose (GE Healthcare). After incubation at 4 °C overnight and several washes, precipitated proteins were eluted with 0.1 M Glycine (pH 3.0) and separated by SDS-PAGE.

## Immunofluorescence

Cells were cultured on glass coverslips for 24 h. After washing with PBS, cells were incubated with 4% paraformaldehyde for 10 min and then permeabilized with 0.2% Triton X-100 for 10 min at room temperature. The cells were then blocked in 5% BSA and incubated with primary antibody at room temperature for 1 h, washed three times with PBST (0.1% Tween 20 in PBS) and incubated with Alexa Fluor 488 or 546 antibody (1:1000, Thermo Fisher Scientific) for 1 h at room temperature. After three washes, all coverslips were mounted with ProLong Gold antifade with DAPI reagent (Thermo Fisher Scientific). Fluorescence images were captured by LSM 710 (Zeiss) confocal microscopy.

## Ubiquitylation assays in cells

To detect ubiquitylation of TEADs in HLF cells, a biotinylated Avi-tagged TEAD (Bio-TEAD) was introduced into HLF cells co-expressing BirA, a bacteria biotin ligase which conjugates biotin to the Avi-tag, a biotin-acceptor peptide using lentivirus expression system. Biotin at 2 μg/mL was added to culture media overnight before cell harvest. Cells were then lysed in 6 M urea buffer. After sonication, lysates were cleared using centrifugation and incubated with Streptavidin-agarose resins overnight at room temperature. Subsequently, the pulldown products were washed five times using 6 M urea buffer. Ubiquitylated TEADs were detected by Western blotting using an anti-ubiquitin antibody. Alternatively, HEK293T cells were co-transfected with Flag-TEADs, Myc-RNF214 and HA-Ub. Cells were lysed in SDS-denaturing buffer (62.5 mM Tris-HCl [pH 6.8], 2% SDS, 10% glycerol) and sonicated. Cleared cell lysates were then diluted 10 to 15-fold in native lysis buffer (50 mM Tris-HCl [pH 7.5], 0.5% Triton X-100, 200 mM NaCl, 10% glycerol). The supernatants were incubated with anti-Flag beads at 4 °C for 2 h. The immunocomplexes were washed five times using native lysis buffer, resolved on SDS-PAGE, and immunoblotted using anti-HA antibody.

For the Halo-ThUBDs assay, we expressed and purified ThUBDs, the ubiquitin affinity matrix[42], which binds selectively to polyubiquitin chains, as Halo-tagged recombinant proteins (Halo-ThUBDs) in BL21(DE3) bacteria cells. Proteins were extracted from HCC cells with 1% Triton lysis buffer containing protease, phosphatase inhibitors and 10 mM N-Ethylmaleimide. A total of 8 μg Halo-ThUBDs recombinant proteins were incubated with 2 mg total lysates from each sample for 3 h at 4 °C. The Halo beads were then washed three times and eluted with SDS sample loading buffer, separated on SDS-PAGE, and detected using Western blotting.

## Pulldown assay

GST-TEAD1 was expressed and purified from BL21 (DE3) bacteria cells. Strep-RNF214 was purified from SF9 insect cell infected by

recombinant baculovirus constructed using Bac-to-Bac™ Baculovirus Expression System (Invitrogen). Proteins bound on beads were mixed with different prey proteins at 4 °C for 2 h in 1% Triton lysis buffer, and then washed five times using the same buffer. The input and pulldown samples were loaded to SDS-PAGE and detected by Ponceau S staining or Western blotting.

For in vitro polyubiquitin chain binding assay, GST-YAP and GST-TAZ were expressed and purified from BL21 (DE3) bacteria cells. Poly-K48 Ubiquitin (3-7) and Poly-K63 Ubiquitin (3-7) were purchased from R&D Systems.

## Animal model

For subcutaneous xenograft model, a total of $2 \times 10^6$ Huh7 cells with indicated treatments were suspended in 100 µl PBS with Matrigel (1:1) and then injected into 5-week-old nude mice. 9 days after injection, the subcutaneous tumors were counted and tumor sizes were measured every 2 days using the Vernier caliper as follows: tumor volume = $(L \times W^2)/2$, where L is the long axis and W is the short. After 21 days of injection, mice were sacrificed and tumors were harvested, weighed and photographed. We used a humane protocol in xenograft tumor growth assay with the endpoints of tumor volume <1500 mm$^3$ permitted by the Animal Ethics Committee of Zhejiang University. The maximal tumor size/burden in this study was not exceeded the limit at the end of the experiments. All mice used were male BALB/c nude mice obtained from Shanghai SLAC Laboratory Animal Company.

For hydrodynamic tail vein injection model, 4-week-old ICR mice were anesthetized by isoflurane, and then plasmids suspended in sterile Ringer's solution (5.6 mM KCl, 154 mM NaCl, 2.2 mM CaCl$_2$, 2.4 mM NaHCO$_3$) in a volume equal to 10% of the body weight were injected in 5–7 s via the tail vein of mice. Plasmids for hydrodynamic injection were prepared using the Qiagen EndoFreeMaxi Kit. The amount of injected DNA was 25 µg *piggyBac* transposase and 41.67 µg of total transposon plasmids. For *Rnf214* knockdown, two shRNAs were designed and expressed by the U6 promoter in tandem with *RAS*. Mice were sacrificed 120 days after injection. Livers were harvested, weighed and photographed. All mice used were male ICR mice and purchased from Shanghai SLAC Laboratory Animal Company. The shRNA sequences against *Rnf214* were provided in Supplementary Table 1.

Standard laboratory chow diet for mice was purchased from XieTong Biology (Cat#1010082) and the SPF grade animal room was maintained with humidity at 45–60% and a 12-h (7:00 a.m. – 7:00 p.m.) light/dark cycle. All animal experiments were approved by the Animal Ethics Committee of Zhejiang University.

## Statistical analysis

Data are presented as the mean ± SD and three levels of significance were presented. Statistical analysis used Student's *t* test, Spearman's correlation analysis, log-rank test and Cox regression analysis with GraphPad Prism software v 7.0 (San Diego, CA. USA). The statistical analysis of the overall survival was done through the Human Protein Atlas website (https://www.proteinatlas.org/). The statistical analysis of the progression free survival was through the Kaplan Meier Plotter website (http://kmplot.com/analysis/). The Spearman's rank correlation analysis of the TCGA cohort in Fig. 5k–m and Supplementary Fig. 5f–l was done through the bioinformatic website (https://www.aclbi.com/static/index.html#/).

## Reporting summary

Further information on research design is available in the Nature Portfolio Reporting Summary linked to this article.

## Data availability

The mass spectrometry data generated in this study have been deposited in the ProteomeXchange Consortium with the dataset identifier PXD052393. Source data are provided with this paper. And the data in this paper also were shared in a Figshare Dataset[83]. The remaining data are available within the Article, Supplementary Information or Source Data file. Source data are provided with this paper.

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

## Acknowledgements

We thank Dr. Shengda Lin for his critical comments on the work. This work was partially supported by grants from the National Natural Science Foundation of China (Nos. 31970734 (J.J.), 32150014 (J.J.), 32022023 (L.S.), 31970726 (B.Z.), 31925010 (F.L.), and 91953121 (F.L.)), National Key Research and Development Program of China (Nos. 2022YFC3401500 (J.J., X.F.), 2021YFA1300100(F.L.), and 2018YFA0108700(F.L.)) and the Fundamental Research Funds for the Central Universities. We are also grateful to our colleagues at the core facility of the Life Sciences Institute for their assistance of picking clones from Invitrogen ORF Clones library.

## Author contributions

M. Lin performed the majority of the experiments, X. Zheng participated in Western blotting and co-IP experiments, J. Yan was involved in the analysis of some clinical samples, F. Huang did biostatistical analysis, Y. Chen and R. Ding helped with protein purifications, J. Wan and L. Zhang performed mass spectrometry, C. Wang and J. Pan helped with animal model, X. Cao and K. Fu prepared some critical reagents, Y. Lou analyzed some clinical data, X.-H. Feng, J. Ji and B. Zhao provided reagents and advice for the study, F. Lan, L. Shen, and X. He supervised mass spectrometry, statistical analysis and clinical sample analysis, respectively. M. Lin prepared the figures, M. Lin, Y. Chen, and J. Jin contributed to literature search, M. Lin, Y. Qiu, and J. Jin cowrote the manuscript. M. Lin, Y. Chen, Y. Qiu, and J. Jin edited the manuscript. J. Jin and Y. Qiu designed and supervised the work. All authors read and approved the final manuscript.

## Competing interests

The authors declare no competing interests.
