## [Peer Review File · Nature Communications]

The RNF214-TEAD-YAP signaling axis promotes hepatocellular carcinoma progression via TEAD ubiquitylationEditorial Note: Parts of this Peer Review File have been redacted as indicated to maintain the confidentiality of unpublished data.

REVIEWER COMMENTS

Reviewer #1 (Remarks to the Author):

In this paper, the authors investigated the role of RNF214, an E3 ubiquitin ligase whose biological function remains unknown, as illustrated by the fact that there is only one publication mentioning it on PubMed as of June 2023. E3 ubiquitin ligases are part of the ubiquitin-proteasome system (UPS), and they are major regulators of protein stability and function. They can alter protein stability by conjugating proteolytic ubiquitin chains (linked for example through lysine 11 or 48 – so called K11 or K48 linked ubiquitin chains), or alter protein function (such as subcellular localization or protein interactions) by conjugating non-proteolytic chains such as the ones linked through lysine 63 (K63). In order to determine the biological function of RNF214, the authors performed a mass spec-based screen to identify protein interactors that could be substrates of this enzyme and found the TEAD family of transcription factors as high confident interactors of RNF214. TEAD proteins are part of the Hippo pathway where they work with the YAP/TAZ transcriptional co-activators to regulate gene expression involved in cell proliferation and migration. By performing biochemical and cell biology assays, the authors found that RNF214 ubiquitinates the TEAD proteins with non-proteolytic ubiquitin chains linked through K63. Mechanistically, this facilitates the interaction between TEAD and YAP/TAZ proteins to trigger the expression of several target genes of the Hippo pathway such as ANKRD1, CTGF and CYR61, contributing to cell proliferation, migration and tumor growth. Thus, the authors propose a model in which RNF214 acts as a positive regulator of Hippo signaling, by ubiquitinating TEAD proteins to facilitate their interaction with YAP/TAZ and trigger an oncogenic transcriptional program. Overall, this is a very interesting study which is relevant to researchers interested in ubiquitin and/or Hippo signaling in the context of malignant proliferation. I found the paper to be well written and easy to follow, and most of the experiments are nicely presented, performed, well controlled, with the resulting data being very clear. However, I have a few suggestions that I believe will help solidify the major findings reported by the authors, and I believe that once these have been addressed it will be a good fit for publication in Nature Communications.

Major points:

- One of the main findings of that paper is the characterization of the E3 ligase RNF214, whose function, substrates and mode of action have never been studied before. A key feature of this enzyme, reported mostly in Figure 3 and Extended data Figure 3, is the fact that it builds K63-linked polyubiquitin chains on the TEAD proteins. While well performed, these experiments show some modest differences which could benefit from a complementary approach. I would strongly recommend the authors to use ubiquitin constructs where all lysine residues but one, are mutated into arginine in order to generate ubiquitin variants that can only form a particular type of chain. This way, they could express a WT, a lysine less (K0), a K48 only or K63 only, along with TEADs, minus or plus RNF214. The expectation would be that a K63 only variant will still lead to increased TEAD ubiquitination, whereas a K0 or K48 only won't. I believe this will further strengthen the observations of the authors, as well as being a great resource for the ubiquitin community to further study the potential linkage specificity of RNF214 itself. However, by looking at the WCL panel of most ubiquitination experiments, it looks like overexpression of RNF214

globally increases cellular ubiquitination, even when ubiquitin mutants were used. This may suggest that RNF214 can do additional ubiquitin linkages and it will be worth talking about it. Some additional points related to Extended data Figure 3 include panel A, which is hard to analyze, so I suggest the authors repeat it. The extended data Figure 3d is difficult to interpret, as usually you would compare protein stability without perturbation to protein stability with E3 ligase overexpression/knock down. I would suggest the authors treat cells with control or RNF214 siRNAs then perform CHX treatment. Finally, figure 3e and f could be a single panel, especially since 3f seems to have a bubble that occurred during transfer which makes it a bit hard to interpret.

- Another interesting finding is the fact that K63-linked polyubiquitin chains built by RNF214 regulate the interaction between TEAD and YAP/TAZ. The fact that recombinant YAP and TAZ can both bind polyubiquitin chains, especially K63 ones, is novel and fits with the model proposed by the authors. However, it only shows that it can bind ubiquitin chains by themselves and not in the context of K63-linked chains on TEAD proteins. Over the years, several ubiquitin binding domains have been reported and it would be extremely useful if the authors could provide evidence that YAP and TAZ contain such domains. The idea would be to map where the recognition happens, generate mutant versions of these domains in both YAP and TAZ, then validate that it is not binding polyubiquitin chains anymore and show that this compromises the regulation of enhancing the binding of YAP/TAZ to TEAD proteins despite RNF214 activity. I realize that this is not a trivial point to address, especially since it is mentioned in the discussion by the authors that this could represent some future work, but it would strengthen the overall findings reported by the authors.

- The identification of ubiquitination sites on TEAD2 and TEAD4, and their subsequent mutation to analyze their impact on protein function, is a great way to validate the importance of non-proteolytic ubiquitination in controlling TEAD protein function. While it seems clear that mutating lysine 345 into arginine (K345R) disturbs ubiquitination and binding to YAP (Figure 4e and f), it seems that there might be compensatory mechanisms and/or additional sites by judging on the data presented in 4g so I think the authors should correct the text accordingly. Moreover, I would recommend the authors to use these single lysine ubiquitin constructs mentioned above to further demonstrate that K345 of TEAD2 is the main acceptor site for K63 chains. The prediction would be that this is completely abrogated by the KR mutation, because in its current form this mutant still gets heavily ubiquitinated, and the differences reported in both figure 4d and e are subtle, even though they go in the direction the authors claim. Some minor points related to figure 4 include panel a, which will benefit from having the RD mutant like figure 4b, and the coIP panel from figure 4f could use a shorter exposure to see the differences even better.

- Another major finding is that RNF214 is a positive regulator of Hippo signaling through its interaction with the TEAD proteins. This is shown by co-immunoprecipitation (coIP) experiments and while they are done very nicely, I would recommend the authors to perform it in a reciprocal manner for TEAD2, 3 and 4, especially for TEAD2 & 3, since they seem to bind non-specifically to the resin that was used in these experiments. I would suggest the authors to only show one in the main figures and the reciprocal in a supplementary panel. Alternatively, they could do something akin to what they show in Supplementary Figure 2g, where reciprocity is shown on a single panel in a very convincing way. Also, these are all done using overexpression approaches, so an endogenous or at least semi endogenous interaction studies should be performed. Related to interaction experiments, the co-IP with YAP would also benefit from the reciprocal approach, and could maybe fit later in the paper? Especially since it raises the question of whether the interaction between RNF214 with YAP

is direct or not, based on the rest of the paper. I believe this could be tested when cells have been knocked down for TEAD or not, and interaction assessed by coIP.

- Related to RNF214 characterization, the effect of the NTD truncation on its interaction with TEAD1 and 2 is a bit hard to follow: while it weakens it, it is certainly not abrogated it. Moreover, there is no appropriate control for Extended data Fig2j, such as Flag-RNF214 alone. The effect observed for the luciferase reporter assay is not as strong as the RD or CCD mutants, and the discrepancy of effect observed for mRNA levels of the different target genes analyzed in Fig2l contributes to the confusion. I would suggest the authors to maybe leave it out of the manuscript, and potentially discuss how it would be worth pursuing in a later work a more thorough study of truncating different parts of RNF214 to better understand the contribution of these different domains. I would also suggest the authors to add the schematic of the different RNF214 constructs they are using in the main figure, as I believe this will facilitate the reading.

- Finally, the authors start their story by mentioning that they generated an RNF214 KO mouse, isolated MEFs and noticed a growth defect, hence pursuing the characterization of RNF214 function for cell fitness. Having MEFs with a WT or RNF214 $-/-$ background constitutes a great system to validate the findings reported all along the study, especially when it comes to the activation of the transcriptional program regulated by the TEAD/YAP/TAZ axis. One could think of performing the luciferase reporter assay in these cells, and also starve these cells and stimulate them with serum in a similar way to what they did in Figure 3c, before following protein expression of Hippo pathway target genes.

Minor points:

- Because nothing is known about RNF214, the authors could perform a simple analysis of its subcellular localization using either cell fractionation and/or immunofluorescence studies.

- In figure 1, what is the "0.1% Input" lane next to each IP panel? To me, the input is the whole cell lysates panel under the IP panels that show expression of the transfected constructs. Please clarify.

- In figure 2, ANKRD1 mRNA levels are not back to basal levels but protein levels are, so that might be worth mentioning. The CTGF blot in Extended data 2a is a bit hard to see and I would recommend the authors to provide a better one. Also, if possible, doing the rescue in figure 2b like what was done in 2a would be a nice addition. In figure 2c, I would suggest the authors blot for phospho-YAP as a surrogate of Hippo pathway activation or not. Finally, in Extended data Fig2f, the HA IP panel seems to be ponceau stained. If that's the case, it should be mentioned in the legend or the figure.

- In figure 4, is the K260R mutation from TEAD4 the equivalent of the K345R mutation of TEAD2? If that is the case, it would be helpful to have it stated clearly in the text.

- In figure 6, I believe panel i has shCK and shRNF214 inverted, same with Extended Fig 6h.

- While the article is extremely well written, I found a few typos all along the manuscript that I am attaching here for the authors to correct.

Line 76: outcomes instead of destinies.

Line 84: alteration instead of alternation.

Line 128: RNF214 function is probably associated with proliferative processes.

Line 183: increase instead of lift

Line 189: and instead of whereas
Line 192: demonstrated that RNF214
Line 229: we decided next
Line 263: wild type and not wide type
Line 276: it is instead of it's
Line 334: implicated instead of implicating
Line 336: mRNA levels are or mRNA level is
Line 356: implicating that RNF214 might
Line 383: statistically significant
Line 452: remove etc
Line 479: By combining clinical data and biological analysis

Reviewer #2 (Remarks to the Author):

The manuscript titled "The RNF214-TEAD-YAP signaling axis promotes hepatocellular carcinoma progression via TEAD ubiquitylation" investigates the functional role of RNF214 in the Hippo pathway and its interaction with TEAD transcription factors. The authors demonstrate that RNF214 is critical for regulating the expression of Hippo pathway target genes through knockdown and overexpression experiments. Mechanistically, RNF214 interacts with TEAD transcription factors and promotes their nonproteolytic polyubiquitylation, acting as a ubiquitin ligase for TEADs and conjugating nonproteolytic K63 polyubiquitin chains on these proteins. These findings establish RNF214 as a novel component involved in the transcriptional regulation of the Hippo pathway. Overall, the manuscript provides valuable insights into the molecular mechanisms underlying the regulation of the Hippo pathway by RNF214, contributing to our understanding of this pathway, and offering potential avenues for further research in the field. While the manuscript is well-written and has broad relevance to cancer and other pathologies involving the Hippo pathway, there are a few minor concerns that need to be addressed before it can be considered for publication.

Specific Comments:

How can RNF214 depletion affect the binding of YAP to TEADs, but not affect the nuclear localization of YAP? In previous studies, the molecular mechanism of YAP's nuclear localization has been extensively investigated. It is widely recognized that while YAP shuttles between the cytoplasm and nucleus, its nuclear retention depends on its interaction with TEAD transcription factors. Disruption of the YAP-TEAD interaction can lead to YAP's cytoplasmic sequestration, resulting in its inactivation. What allows YAP to retain nuclear localization in the absence of TEAD interaction? Are there other mechanisms or factors that compensate for the loss of TEAD binding by YAP?

Also, while the in vitro data is convincing, those findings cannot be extrapolated to liver cancer. Although the authors use cancer cell lines in subcutaneous xenograft models, those models do not recapitulate tumor initiation nor tumor maintenance in the liver. Not only due to the abnormal location for tumor growth outside the liver parenchyma, but also because xenograft models lack the key contribution of the immune system. The authors could easily use hydrodynamic tail vein injection of sleeping beauty constructs to perform gain and loss of function experiments in liver cancer. This will provide more relevant insights into the impact of RNF214 in liver cancer. This model would allow the researchers to demonstrate

the function of RND214 in tumors driven by YAP or other oncogenes. In addition, the authors could explore its function in tumor initiation vs tumor maintenance.

Our answers to reviewers' comments:

Our answers to Reviewer 1's comments:

In this paper, the authors investigated the role of RNF214, an E3 ubiquitin ligase whose biological function remains unknown, as illustrated by the fact that there is only one publication mentioning it on PubMed as of June 2023. E3 ubiquitin ligases are part of the ubiquitin-proteasome system (UPS), and they are major regulators of protein stability and function. They can alter protein stability by conjugating proteolytic ubiquitin chains (linked for example through lysine 11 or 48 – so called K11 or K48 linked ubiquitin chains), or alter protein function (such as subcellular localization or protein interactions) by conjugating non-proteolytic chains such as the ones linked through lysine 63 (K63). In order to determine the biological function of RNF214, the authors performed a mass spec-based screen to identify protein interactors that could be substrates of this enzyme and found the TEAD family of transcription factors as high confident interactors of RNF214. TEAD proteins are part of the Hippo pathway where they work with the YAP/TAZ transcriptional co-activators to regulate gene expression involved in cell proliferation and migration. By performing biochemical and cell biology assays, the authors found that RNF214 ubiquitinates the TEAD proteins with non-proteolytic ubiquitin chains linked through K63. Mechanistically, this facilitates the interaction between TEAD and YAP/TAZ proteins to trigger the expression of several target genes of the Hippo pathway such as ANKRD1, CTGF and CYR61, contributing to cell proliferation, migration and tumor growth. Thus, the authors propose a model in which RNF214 acts as a positive regulator of Hippo signaling, by ubiquitinating TEAD proteins to facilitate their interaction with YAP/TAZ and trigger an oncogenic transcriptional program.

Overall, this is a very interesting study which is relevant to researchers interested in ubiquitin and/or Hippo signaling in the context of malignant proliferation. I found the paper to be well written and easy to follow, and most of the experiments are nicely presented, performed, well controlled, with the resulting data being very clear. However, I have a few suggestions that I believe will help solidify the major findings reported by the authors, and I believe that once these have been addressed it will be a good fit for publication in Nature Communications.

Answer:

We thank Reviewer 1 for the very positive comments on our study and constructive questions about our manuscript. We have revised the manuscript according to specific comments as described below.

Major points:

- One of the main findings of that paper is the characterization of the E3 ligase RNF214, whose function, substrates and mode of action have never been studied before. A key feature of this enzyme, reported mostly in Figure 3 and Extended data Figure 3, is the fact that it builds K63-linked polyubiquitin chains on the TEAD proteins. While well performed, these experiments show some modest differences which could benefit from a complementary approach. I would strongly recommend the authors to use ubiquitin constructs where all lysine residues but one, are mutated into arginine in order to generate

ubiquitin variants that can only form a particular type of chain. This way, they could express a WT, a lysine less (K0), a K48 only or K63 only, along with TEADs, minus or plus RNF214. The expectation would be that a K63 only variant will still lead to increased TEAD ubiquitination, whereas a K0 or K48 only won't. I believe this will further strengthen the observations of the authors, as well as being a great resource for the ubiquitin community to further study the potential linkage specificity of RNF214 itself. However, by looking at the WCL panel of most ubiquitination experiments, it looks like overexpression of RNF214 globally increases cellular ubiquitination, even when ubiquitin mutants were used. This may suggest that RNF214 can do additional ubiquitin linkages and it will be worth talking about it. Some additional points related to Extended data Figure 3 include panel A, which is hard to analyze, so I suggest the authors repeat it. The extended data Figure 3d is difficult to interpret, as usually you would compare protein stability without perturbation to protein stability with E3 ligase overexpression/knock down. I would suggest the authors treat cells with control or RNF214 siRNAs then perform CHX treatment. Finally, figure 3e and f could be a single panel, especially since 3f seems to have a bubble that occurred during transfer which makes it a bit hard to interpret.

Answer:

We thank for Reviewer 1's comments and suggestions. We have carried out new experiments showing that RNF214 does not regulate the protein stability of TEADs:

1. RNF214 knockdown using siRNA oligos in Hep3b cells or knockout using the CRISPR/Cas9 method in HLF cells did not alter protein stabilities of TEADs (Fig.3f and Supplemental Fig.3d in the revised manuscript).
2. Overexpression of RNF214 did not affect protein stabilities of TEADs in HEK293A cells treated with cycloheximide (Supplemental Fig.3e in the revised manuscript).

In our first submission, we have shown that RNF214 promotes TEADs ubiquitylation using the BirA-Avi system. We have carried out two new experiments to replace the old supplemental Figure 3a which is a little bit hard to analyze:

- 1 . RNF214 knockdown in HLF cells significantly decreased TEAD2 ubiquitylation (Fig.3d of the revised manuscript).
- 2 . Overexpression of RNF214 promoted TEAD2 ubiquitylation in Huh1 cells, another HCC cell line. (Supplemental Fig.3a of the revised manuscript).

Furthermore, as the Reviewer 1 suggested, we generated a lysine less (K0) and a panel of K-only ubiquitin mutants. We then co-expressed Flag-TEAD2, and wild type or K-only ubiquitin constructs in HEK293T cells. Overall, we found the wild type ubiquitin supported the strongest ubiquitylation of TEAD2. Several K-only mutants, including K6, K33, especially K27 behaved better than the other ubiquitin mutants. Like our previous data, we observed that the K63 only mutant is better than the K48 only one (Fig.3g of the revised manuscript). We did the similar experiment using the KR mutants of HA-Ub, and found that the K0 mutant inhibited TEAD2 ubiquitylation at the most and the K27R mutant behaved similarly (Fig.3h of the revised manuscript). More importantly, we observed that the K27-only mutant supported the Myc-RNF214-enhanced TEAD2 ubiquitylation as good as the wild type did (Fig.3i of the revised manuscript), further indicating that RNF214 might conjugate non-proteolytic K27 polyubiquitin chains on TEADs. However, we cannot rule out the possibility that RNF214 synthesizes mixed polyubiquitin chains on TEADs.

- Another interesting finding is the fact that K63-linked polyubiquitin chains built by RNF214 regulate the interaction between TEAD and YAP/TAZ. The fact that recombinant YAP and TAZ can both bind polyubiquitin chains, especially K63 ones, is novel and fits with the model proposed by the authors. However, it only shows that it can bind ubiquitin chains by themselves and not in the context of K63-linked chains on TEAD proteins. Over the years, several ubiquitin binding domains have been reported and it would be extremely useful if the authors could provide evidence that YAP and TAZ contain such domains. The idea would be to map where the recognition happens, generate mutant versions of these domains in both YAP and TAZ, then validate that it is not binding polyubiquitin chains anymore and show that this compromises the regulation of enhancing the binding of YAP/TAZ to TEAD proteins despite RNF214 activity. I realize that this is not a trivial point to address, especially since it is mentioned in the discussion by the authors that this could represent some future work, but it would strengthen the overall findings reported by the authors.

Answer:

We thank for Reviewer 1's insight comments. We also really appreciate Reviewer 1's understanding that mapping and proving a new ubiquitin binding domain need lots of new data, perhaps even structural data. We have compared the common domains between YAP and TAZ (Figure 1 for reviewer), generated a few mutants and carried out some new experiments. The outcome is very interesting. However, we hope to hold these new data for a future publication:

1. We have done *in vitro* GST pulldown assays using all eight tetra-Ub chains as baits. We found that both GST-YAP and GST-TAZ directly bind to M1, K27 and K63 chains significantly (Figure 2 for reviewer). It is worth noting that YAP and TAZ have the strongest ability to bind tetra-K27 Ub chains. As compared with our previous data in which only included the K48 and K63 Ub chains, the current data are more complete.
2. According to the common domains of YAP and TAZ, we have constructed several deletion mutants. The WW domain is common to YAP and TAZ and there are two WW domains in YAP, but only one in TAZ. In Fig.4j-k of our first submission, YAP binds stronger to poly-Ub chains than TAZ. So, we first hypothesized that the WW domain might be important for YAP/TAZ's binding to Ub chains. Therefore, we have made three additional mutants surrounding WW domain (Figure 3 for reviewer). Using *in vitro* GST pulldown assays, we surprisingly found the domains beyond WW domain may contribute to the binding, since both GST-YAP Δ C and GST-TAZ Δ C failed to bind poly-K63 Ub chains like the full length (Figure 4 for reviewer). Then, we made two more additional YAP mutants, including Δ NTD (deletion of N-terminal proline rich domain) and Δ TBD (deletion of C-terminal TEAD binding domain). The Δ TBD mutant did not affect the binding, which reminds us that the region between WW domain and TEAD binding domain may act as a potential ubiquitin binding domain.
3. Finally, we did an *in vitro* GST pulldown assay using the tetra-K27 Ub chain. Consistent with the results mentioned above, the GST-YAP Δ C failed to bind the

tetra-K27 Ub chain (Figure 6 for reviewer).

Together, we have found that the region between the WW domain and the TEAD binding domain, including the STAT1 binding domain and the coiled-coil domain, is essential for YAP/TAZ to interact with poly-ubiquitin chains. Our data also suggested that YAP could bind to different linkages of polyubiquitin chains. We are performing more experiments to refine the sequences which are responsible for YAP/TAZ to recognize polyubiquitin chains.

[Redacted]

Figure 1 for reviewer. A schema of the structures of YAP and TAZ.

[Redacted]

Figure 2 for reviewer. *In vitro* tetra-Ub chain pulldown assay of YAP and TAZ. GST-YAP, GST-TAZ and tetra-M1 chain were purified from BL21 (DE3) bacteria cells; tetra-K6 Ub chains and tetra-K11 Ub chains were purchased from R&D Systems; others were provided by our collaborator.

[Redacted]

Figure 3 for reviewer. A schema of the mutants of YAP and TAZ.

[Redacted]

Figure 4 for reviewer. *In vitro* poly-K63 Ub chain pulldown assay of YAP and TAZ mutants. GST-YAP or GST-TAZ mutants were purified from BL21 (DE3) bacteria cells and poly-K63 Ub (3-7) chain was purchased from R&D Systems.

[Redacted]

Figure 5 for reviewer. *In vitro* poly-K63 Ub chain pulldown assay of YAP mutants. GST-YAP mutants were purified from BL21 (DE3) bacteria cells and poly-K63 Ub (3-7) chain was purchased from R&D Systems.

[Redacted]

Figure 6 for reviewer. *In vitro* tetra-K27 Ub chain pulldown assay of YAP mutants. GST-YAP mutants were purified from BL21 (DE3) bacteria cells and tetra-K27 Ub chain were gently provided by our collaborator.

- The identification of ubiquitination sites on TEAD2 and TEAD4, and their subsequent mutation to analyze their impact on protein function, is a great way to validate the importance of non-proteolytic ubiquitination in controlling TEAD protein function. While it

seems clear that mutating lysine 345 into arginine (K345R) disturbs ubiquitination and binding to YAP (Figure 4e and f), it seems that there might be compensatory mechanisms and/or additional sites by judging on the data presented in 4g so I think the authors should correct the text accordingly. Moreover, I would recommend the authors to use these single lysine ubiquitin constructs mentioned above to further demonstrate that K345 of TEAD2 is the main acceptor site for K63 chains. The prediction would be that this is completely abrogated by the KR mutation, because in its current form this mutant still gets heavily ubiquitinated, and the differences reported in both figure 4d and e are subtle, even though they go in the direction the authors claim. Some minor points related to figure 4 include panel a, which will benefit from having the RD mutant like figure 4b, and the colP panel from figure 4f could use a shorter exposure to see the differences even better.

Answer:

We thank for Reviewer 1's comments and suggestions. We agree that the K345R residue of TEAD2 is the major ubiquitylation site mediated by RNF214 and we have modified the text accordingly (page 12, line 3). Besides, we have modified experiments of the figure 4a (in the first submission) that the RNF214 RD mutant failed to promote the interaction between YAP and TEAD2 (Fig.4a of the revised manuscript). And we showed a shorter exposure to see the differences between TEAD2 WT and K345R mutant (Fig. 4f of the revised manuscript).

- Another major finding is that RNF214 is a positive regulator of Hippo signaling through its interaction with the TEAD proteins. This is shown by co-immunoprecipitation (coIP) experiments and while they are done very nicely, I would recommend the authors to perform it in a reciprocal manner for TEAD2, 3 and 4, especially for TEAD2 & 3, since they seem to bind non-specifically to the resin that was used in these experiments. I would suggest the authors to only show one in the main figures and the reciprocal in a supplementary panel. Alternatively, they could do something akin to what they show in Supplementary Figure 2g, where reciprocity is shown on a single panel in a very convincing way. Also, these are all done using overexpression approaches, so an endogenous or at least semi endogenous interaction studies should be performed. Related to interaction experiments, the co-IP with YAP would also benefit from the reciprocal approach, and could maybe fit later in the paper? Especially since it raises the question of whether the interaction between RNF214 with YAP is direct or not, based on the rest of the paper. I believe this could be tested when cells have been knocked down for TEAD or not, and interaction assessed by coIP.

Answer:

We really appreciate Reviewer 1's comments and suggestions. Following this reviewer's suggestions, we have done reciprocal co-IP experiments and confirmed the interactions between RNF214 and TEAD3 or TEAD4 (Fig.1f and 1g in the revised manuscript). Moreover, we have found better data of interactions between RNF214 and TEAD2 with lower non-specific background than previous data (Supplemental Fig.1d in the first submission; the better data was shown in the Supplementary Fig.1d of the revised manuscript). Second, using semi-endogenous experiments, we observed Flag-RNF214

interacts with endogenous pan-TEAD and TEAD2 in HEK293A cells (Fig. 1h of the revised manuscript), as well as in Hep3b cells (Supplemental Fig.1f of the revised manuscript). We also validated the TEAD2 band using siRNA oligo in HEK293A cells (Fig.7 for reviewer, and also in the Supplementary Fig.1e in the revision). Third, we confirmed the interaction between RNF214 and TEAD2 at endogenous levels in the HCC cell lines (Fig. 1i and Supplemental Fig.1f of the revised manuscript). We also agree with Reviewer 1 that the data of the interaction between RNF214 with YAP fit latter in the paper, therefore, we have moved these data from Fig.1k of the first submission to Supplementary Fig.4f of the revised manuscript. Our data indicated that the interaction between RNF214 with YAP is indirect since they could not interact with each other in the TEAD1/3/4 knock-down cells (Supplemental Fig.1g of the revised manuscript). All together, these results confirmed RNF214 associates with TEADs and plays important roles in regulating transcriptional activities of the YAP-TEAD complex.

Figure 7 for reviewer. Knock-down of TEAD2 in HEK293A cells.

- Related to RNF214 characterization, the effect of the NTD truncation on its interaction with TEAD1 and 2 is a bit hard to follow: while it weakens it, it is certainly not abrogated it. Moreover, there is no appropriate control for Extended data Fig2j, such as Flag-RNF214 alone. The effect observed for the luciferase reporter assay is not as strong as the RD or CCD mutants, and the discrepancy of effect observed for mRNA levels of the different target genes analyzed in Fig2l contributes to the confusion. I would suggest the authors to maybe leave it out of the manuscript, and potentially discuss how it would be worth pursuing in a later work a more thorough study of truncating different parts of RNF214 to better understand the contribution of these different domains. I would also suggest the authors to add the schematic of the different RNF214 constructs they are using in the main figure, as I believe this will facilitate the reading.

Answer:

We really appreciate Reviewer 1's wonderful suggestions. Accordingly, we have added a schema of the different RNF214 constructs in the main figure (Fig.2g of the revised manuscript) and agreed to leave the NTD truncation out of the manuscript, including Fig.2k-l and Supplementary Fig.2i-k of the first submission. We also suggested that RNF214 potentially recognize TEADs via its NTD domain in the discussion of the revised manuscript (Page 17, line 10).

- Finally, the authors start their story by mentioning that they generated an RNF214 KO mouse, isolated MEFs and noticed a growth defect, hence pursuing the characterization of RNF214 function for cell fitness. Having MEFs with a WT or RNF214 ^{-/-} background constitutes a great system to validate the findings reported all along the study, especially when it comes to the activation of the transcriptional program regulated by the TEAD/YAP/TAZ axis. One could think of performing the luciferase reporter assay in these cells, and also starve these cells and stimulate them with serum in a similar way to what they did in Figure 3c, before following protein expression of Hippo pathway target genes.

Answer:

We really appreciate Reviewer 1's wonderful suggestions. As the reviewer suggested, we have compared the functions of the TEAD/YAP/TAZ transcription complex between the WT and the RNF214^{-/-} MEF cells. First, we directly compared expression levels of the target genes of YAP/TAZ-TEADs complex between two independent pairs of the WT and RNF214^{-/-} MEFs which were generated from two litters. We noticed that ANKRD1 and CTGF were downregulated in RNF214^{-/-} MEFs (Supplementary Fig. 2c of the revised manuscript). Second, we starved these MEFs and stimulated them with serum. We reconfirmed that knocking out RNF214 inhibited expressions of target genes of YAP/TAZ-TEADs induced by serum in MEFs (Fig.2d of the revised manuscript). These new data strongly support that RNF214 participates in regulating the transcriptional activities of the YAP/TAZ-TEADs complex.

Minor points:

- Because nothing is known about RNF214, the authors could perform a simple analysis of its subcellular localization using either cell fractionation and/or immunofluorescence studies.

Answer:

Indeed, we found RNF214 localized both in the cytoplasm and the nucleus using the IF method (Supplemental Fig.1a of the revised manuscript).

- In figure 1, what is the "0.1% Input" lane next to each IP panel? To me, the input is the whole cell lysates panel under the IP panels that show expression of the transfected constructs. Please clarify.

Answer:

In our co-IP experiments, "0.1% Input" means 0.1% of whole cell lysates which were used for IP. We now indicated this in the figure legends.

- In figure 2, ANKRD1 mRNA levels are not back to basal levels but protein levels are, so that might be worth mentioning. The CTGF blot in Extended data 2a is a bit hard to see and I would recommend the authors to provide a better one. Also, if possible, doing the rescue in figure 2b like what was done in 2a would be a nice addition. In figure 2c, I would suggest the authors blot for phospho-YAP as a surrogate of Hippo pathway activation or not. Finally, in Extended data Fig2f, the HA IP panel seems to be ponceau stained. If that's the case, it should be mentioned in the legend or the figure.

Answer:

We appreciate Reviewer 1's suggestions. First, the mRNA level of ANKRD1 was not equal to protein level, so there might be other potential mechanisms. We had discussed this in the revised manuscript (page 6, line 22). Second, we did the rescue experiments in Huh7 cells (Fig.2b and Supplementary Fig.2b of the revised manuscript) and provided a better CTGF blot for the rescue experiment in Hep3b cells (Supplementary Fig. 2a of the revised manuscript).

As suggested, we blotted the phospho-YAP as indication of Hippo-off under serum stimulation (Fig. 2c of the revised manuscript).

Lastly, the HA IP panel in the previous Fig. 2f was ponceau staining and we had made a note in the figure and also mentioned it in the Figure legend (Supplementary Fig.2i of the revised manuscript).

- In figure 4, is the K260R mutation from TEAD4 the equivalent of the K345R mutation of TEAD2? If that is the case, it would be helpful to have it stated clearly in the text.

Answer:

Yes, the K260R mutation from TEAD4 is equal to the K345R mutation of TEAD2 and we have now stated it clearly in the revised manuscript (page 11, the last line).

- In figure 6, I believe panel i has shCK and shRNF214 inverted, same with Extended Fig 6h.

Answer:

We thank for Reviewer 1's comments and apologize for the mistake. We did invert the control group and experiment group and have corrected it in the figure (Fig. 6e and Supplementary Fig.7f of the revised manuscript).

- While the article is extremely well written, I found a few typos all along the manuscript that I am attaching here for the authors to correct.

Line 76: outcomes instead of destinies.

Line 84: alteration instead of alternation.

Line 128: RNF214 function is probably associated with proliferative processes.

Line 183: increase instead of lift

Line 189: and instead of whereas

Line 192: demonstrated that RNF214

Line 229: we decided next

Line 263: wild type and not wide type

Line 276: it is instead of it's

Line 334: implicated instead of implicating

Line 336: mRNA levels are or mRNA level is

Line 356: implicating that RNF214 might

Line 383: statistically significant

Line 452: remove etc

Line 479: By combining clinical data and biological analysis

Answer:

We thank for Reviewer 1's suggestions and have corrected these grammar mistakes

in the revision as highlighted.

Our answers to Reviewer 2's comments:

Reviewer #2 (Remarks to the Author):

The manuscript titled "The RNF214-TEAD-YAP signaling axis promotes hepatocellular carcinoma progression via TEAD ubiquitylation" investigates the functional role of RNF214 in the Hippo pathway and its interaction with TEAD transcription factors. The authors demonstrate that RNF214 is critical for regulating the expression of Hippo pathway target genes through knockdown and overexpression experiments. Mechanistically, RNF214 interacts with TEAD transcription factors and promotes their nonproteolytic polyubiquitylation, acting as a ubiquitin ligase for TEADs and conjugating nonproteolytic K63 polyubiquitin chains on these proteins. These findings establish RNF214 as a novel component involved in the transcriptional regulation of the Hippo pathway. Overall, the manuscript provides valuable insights into the molecular mechanisms underlying the regulation of the Hippo pathway by RNF214, contributing to our understanding of this pathway, and offering potential avenues for further research in the field. While the manuscript is well-written and has broad relevance to cancer and other pathologies involving the Hippo pathway, there are a few minor concerns that need to be addressed before it can be considered for publication.

Answer:

We thank for Reviewer 2's positive comments and constructive suggestions. We answered the reviewer's specific questions as follows

Specific Comments:

How can RNF214 depletion affect the binding of YAP to TEADs, but not affect the nuclear localization of YAP? In previous studies, the molecular mechanism of YAP's nuclear localization has been extensively investigated. It is widely recognized that while YAP shuttles between the cytoplasm and nucleus, its nuclear retention depends on its interaction with TEAD transcription factors. Disruption of the YAP-TEAD interaction can lead to YAP's cytoplasmic sequestration, resulting in its inactivation. What allows YAP to retain nuclear localization in the absence of TEAD interaction? Are there other mechanisms or factors that compensate for the loss of TEAD binding by YAP?

Answer:

We really appreciate Reviewer 2's insight comments. Our data showed that RNF214 enhances but is not essential for the interactions of YAP-TEADs. Mechanistically, RNF214 could enhance interactions between YAP and TEADs by ubiquitylating TEADs. On one hand, RNF214 knockdown in HCC cells (Fig.2a-b and Supplementary Fig.2a-b of the revised manuscript) or knockout of RNF214 in MEFs (Fig. 2d and Supplementary Fig. 2c of the revised manuscript) could not completely inhibit the expressions of the downstream target genes; On the other hand, TEADs KR mutant still maintained the basal interaction with YAP as good as WT did (Fig.4f of the revised manuscript) and produced a certain degree of luciferase activities, when YAP was co-expressed (Fig. 4g-h of the revised manuscript).

Based on our data, we could not find any evidence that RNF214 affects localizations of YAP and TEADs or the cytoplasmic-nuclear shuttling of YAP (Supplemental Fig. 4a-c of the revised manuscript).

Also, while the in vitro data is convincing, those findings cannot be extrapolated to liver cancer. Although the authors use cancer cell lines in subcutaneous xenograft models, those models do not recapitulate tumor initiation nor tumor maintenance in the liver. Not only due to the abnormal location for tumor growth outside the liver parenchyma, but also because xenograft models lack the key contribution of the immune system. The authors could easily use hydrodynamic tail vein injection of sleeping beauty constructs to perform gain and loss of function experiments in liver cancer. This will provide more relevant insights into the impact of RNF214 in liver cancer. This model would allow the researchers to demonstrate the function of RNF214 in tumors driven by YAP or other oncogenes. In addition, the authors could explore its function in tumor initiation vs tumor maintenance.

Answer:

We thank for Reviewer 2's wonderful comments. As reported previously, YAP wild type or YAP-S127A alone was not tumorigenic, but YAP-5SA is highly malignant (Jixin Dong et. al., *Cell*, 2007; Yi Liu-Chittenden et. al., *Genes & Development*, 2012; Shuying Shen et. al., *Cell research*, 2015;). It has been reported that aberrant activation of β -Catenin is tightly associated with YAP-related tumorigenesis (Joseph Rosenbluh et. al., *Cell*, 2012; Junyan Tao et. al., *Gastroenterology*, 2014). Based on transcriptomic and proteomic profiles from 25 mouse models of primary liver cancer through hydrodynamic tail vein injection of sleeping beauty constructs, we noticed *Ctnnb1+Nras* combination drastically induced tumorigenesis in more than 90% mice and had a feature of Hippo-off (Mei Tang et. al., *Science Advances*, 2022). Therefore, we did hydrodynamic tail vein injection to induce high expression of *Ctnnb1+Nras* in combination with RNF214 knockdown in the mouse liver to further demonstrate the function of RNF214. Three plasmids were used in the Sleeping Beauty system to co-express *Ctnnb1* and *Nras* together with two *Rnf214* shRNAs (Fig.6f). Overexpression of *Ctnnb1* and *Nras* promoted tumorigenesis in the mouse liver, and *Rnf214* knockdown suppressed tumor formation significantly (Fig.6g-h and Supplementary Fig.7i of the revised manuscript). More importantly, we found the expressions of *Ctgf* and *Cyr61* were obviously lower in *Rnf214* knockdown tumor sections than control tumors (Fig.6i of the revised manuscript). Altogether, this model established the function of RNF214 in liver tumorigenesis.

REVIEWERS' COMMENTS

Reviewer #1 (Remarks to the Author):

The authors have addressed all of my comments and the ones of the other reviewer in a convincing way. The preliminary data they attached on how YAP/TAZ binds to ubiquitin chains is exciting, and will for sure make a nice follow up study to this paper. I highly recommend acceptance of this manuscript as I believe it is now stronger than its initial submission, and definitely a good fit for Nature Communications.

Reviewer #2 (Remarks to the Author):

The authors have satisfactorily answered all of my concerns.